# RoCK and ROI: single-cell transcriptomics with multiplexed enrichment of selected transcripts and region-specific sequencing

Giulia Moro [1,11], Izaskun Mallona [1,2,11] ✉, Malwine J. Barz[3,4], Joël Maillard[1], Michael David Brügger[1], Hassan Fazilaty [1], Quentin Szabo [1], Tomas Valenta [1,5], Kristina Handler[6], Fiona Kerlin[7,8], Lorenz Bastian [3,4], Claudia D. Baldus [3,4], Andreas E. Moor [9], Robert Zinzen [10], Mark D. Robinson [1,2], Erich Brunner [1] ✉ & Konrad Basler[1]

Single-cell profiling technologies allow exploring molecular mechanisms that drive development, health, and disease. However, current methods still fall short of profiling single cell transcriptomes comprehensively, with one major challenge being high non-detection rates of specific transcripts and transcript regions. Such information is often crucial to understanding the biology of cells. Here, we introduce RoCK and ROI (Robust Capture of Key transcripts and Regions Of Interest), a scRNA-seq workflow encompassing two techniques. RoCKseq uses targeted capture to enrich for key transcripts, thereby supporting the detection and identification of cell types and complex phenotypes in scRNA-seq experiments. ROIseq directs a subset of reads to a specific region of interest via selective priming. Importantly, RoCK and ROI enables retrieval of specific sequence information without compromising overall single cell transcriptome information. We validate RoCK and ROI across diverse biological systems highlighting the versatility and showing the power of the method to retrieve critical transcriptomic features.

Single cell RNA sequencing (scRNA-seq) is a valuable tool to study gene expression in complex and heterogeneous tissues. Main advances that followed the advent of scRNA-seq[1] are the ability to barcode RNA from individual cells[2] and the use of barcoded beads to simultaneously analyze many cells[3]. The beads harbor oligonucleotides (oligos) that are covalently attached to and unique to each bead. Through the combination of a single cell with a uniquely barcoded bead in small reaction chambers, the transcripts of a cell can be captured, discernibly barcoded, individually marked (with unique molecular identifiers, UMIs) and processed into cDNA libraries that are suitable for high throughput sequencing (HTS)[3,4]. These advances have fueled the development of various scRNA-seq technologies that allow in-depth

[1]Department of Molecular Life Sciences, University of Zurich, Zurich, Switzerland. [2]SIB Swiss Institute of Bioinformatics, University of Zurich, Zurich, Switzerland. [3]Medical Department II, Hematology and Oncology, University Hospital Schleswig-Holstein, Kiel, Germany. [4]Clinical Research Unit "CATCH ALL" (KFO 5010), Kiel, Germany. [5]Laboratory of Cell and Developmental Biology, Institute of Molecular Genetics of the Czech Academy of Sciences, Prague, Czech Republic. [6]Institute of Experimental Immunology, University of Zurich, Zurich, Switzerland. [7]Berlin Institute for Medical Systems Biology (BIMSB), Max Delbrück Center for Molecular Medicine (MDC) in the Helmholtz Association, Berlin, Germany. [8]Institute of Biology, Department of Biology, Chemistry, Pharmacy, Free University Berlin, Berlin, Germany. [9]Department of Biosystems Science and Engineering, ETH Zürich, Basel, Switzerland. [10]Systems Biology Imaging Technology Platform, Berlin Institute for Medical Systems Biology (BIMSB), Max Delbrück Center for Molecular Medicine (MDC) in the Helmholtz Association, Berlin, Germany. [11]These authors contributed equally: Giulia Moro, Izaskun Mallona. ✉e-mail: izaskun.mallona@gmail.com; erich.brunner@mls.uzh.ch

transcriptional profiling of selected cell populations and acquisition of multimodal datasets for tissue-derived cell mixtures across health and disease[5].

However, many bead-based high-throughput technologies still suffer from severe limitations. First, the data acquired in such experiments tends to cover a small fraction of each cell's transcriptome[6–8]. The loss of information may occur at various levels, including RNA capture on barcoded beads, reverse transcription, preferential PCR amplification during library generation or sequencing bias[9–13]. As a result, a high proportion of expressed genes remain undetected (i.e., a zero measurement)[8,14,15]. In particular, the detection sensitivity for lowly expressed transcripts remains challenging[8,14].

Previous methods aiming to mitigate the detection limits of transcripts of interest can be subdivided into bioinformatic strategies, including handling data as pseudobulks[16,17], or meta-cells[18]; and so-called targeted wet-lab methods[19–26]. A wide variety of strategies are available targeting the transcripts at different levels. Namely, targeted amplification methods[27,28] require that the transcripts of interest have previously been captured on the beads and reverse transcribed, and therefore do not address the a priori problem of capturing rare transcripts that would be most efficiently done directly at the RNA level. Similar to targeted amplification, other protocols offer enrichment of transcripts of interest via specific probes at the level of library generation (i.e. after first strand synthesis)[29,30] or aim to remove non-informative highly expressed transcripts[31,32]. These methods improve detection of transcripts of interest at the level of library generation and sequencing, but none addresses the fact that loss of information may already occur during RNA capture; only targeted capture of transcripts of interest would solve this issue. One solution, DARTseq[33], used a subset of DNA oligos on barcoded beads that are equipped with nucleotide sequences allowing targeted capture of transcripts. A variable bead modification rate between 25 and 40% was reached, which is reflected by a similar variation of information in the transcriptome profile. Importantly, DARTseq allows both the recovery of transcripts of interest as well as profiles for the transcriptome of cells.

An additional limitation of many scRNA-seq methods is the bias toward 3′ or 5′ readouts of dT-captured transcripts[3,34] resulting in a lower coverage of other regions within the coding sequence (CDS). However, there is often important information in the CDS of a transcript that, if read out, could enhance the value of scRNA-seq experiments. This information may be restricted to short regions of interest (ROIs) in a transcript as is the case for splice junctions or single nucleotide variants. In both cases, reads would need to encompass small but specific regions that are unlikely to be efficiently captured by end-directed sequencing. One way to obtain the sequence information of a short ROI is to use full-length sequencing methods[19,22,34–36] or VASAseq[31], which is based on fragmentation of all RNA molecules in a cell followed by polyadenylation, providing information across the full transcript. Other solutions aiming to detect ROIs in transcripts rely on specific primers or additional amplification steps[20,26,37–40], all of which significantly increase complexity of library generation protocols. Furthermore, these approaches have in common that they use pre-amplified cDNA or synthesized first strands of dT-captured transcripts to amplify the ROI.

Although the described methods increase the detection of transcripts and tackle the technical challenges of sequencing through a ROI, they also come with limitations such as lengthy library generation protocols, increased sequencing cost or the inability to target multiple regions of interest.

Here we present RoCK and ROI, a simple scRNA-seq method that combines three important features in one workflow. Through comparison of RoCK and ROI to available targeted scRNA-seq methods, we highlight that RoCK and ROI is the only method combining targeted capture directly at the RNA level (feature one) with the specific detection of features of interest by HTS (feature two). Third, a standard whole transcriptome analysis (WTA) library is generated for the same cells without sacrificing detection depth, allowing cell-by-cell pairing of WTA, RoCKseq and ROIseq information. RoCK and ROI can achieve robust target detection in up to 98% of cells. Besides presenting thorough validations of RoCK and ROI and the simultaneously-profiled transcriptome, we showcase multiple applications, including splice junction and alternative splicing detection in murine cells, as well as distinguishing between major and minor *BCR::ABL*1 fusion transcripts in two cell lines and a patient sample in the context of cancer. In addition, we provide extensive step-by step documentation describing the wet-lab protocol and the data analysis workflow, supporting easy implementation and usage of the method in other laboratories. We anticipate that RoCK and ROI will be applicable to a wide variety of biological samples, providing new insights into biological mechanisms at the single cell level.

## Results
### RoCKseq bead modification is reproducible, titratable and stable

In order to detect specific transcripts in single cells, we established a method that captures RNAs not via their polyA tail but rather through hybridization to an upstream target site such as within the coding sequence (CDS; see Fig. 1a and Supplementary Fig. 1a). The method uses barcoded beads commercially available for the BD Rhapsody scRNA-seq platform, for which the sequence information of the bead-attached oligos is publicly accessible (Supplementary Fig. 1b). The beads carry two types of oligos: (1) dT oligos, which are needed to capture polyadenylated RNAs to obtain WTA information; and (2) template switching oligos (TSO), standardly used for the VDJ full length (TCR/BCR) assay. To specifically capture RNAs of interest, we reasoned that it should be possible to append the TSO oligos with a capture sequence complementary to the target(s) of interest (referred to as RoCKseq beads). Additionally, by only modifying the TSO-portion we would not compromise the beads' ability to generate WTA libraries. To establish the method, we first focused on appending a single capture sequence complementary to the *eGFP* CDS to the TSOs (Supplementary Fig. 2a, b). The addition of the capture sequence is mediated by a DNA polymerase-based enzymatic reaction using a single stranded DNA oligo (splint) for modification (Fig. 1b and Supplementary Fig. 2a, b). After annealing of the splint to the TSOs, the T4 DNA polymerase elongates the recessed ends, generating double stranded DNA molecules. We chose this enzyme due to its low error rate and inactivation ability by heat, which is essential for subsequent steps in the bead modification protocol.

Since the T4 DNA polymerase has an intrinsic 3′−5′ exonuclease activity that targets single-stranded DNA (ssDNA)[41,42], a phosphorylated polyA oligo is added to protect the dT oligos at this step. Importantly, to restore the modified TSO and dT oligos to the single-strandedness needed for RNA capture, we use a lambda exonuclease to remove the complementary strand. This enzyme strongly prefers phosphorylated 5′-ends compared to unphosphorylated DNA[43,44], hence the addition of the 5′ phosphate groups to the splint and protective oligos.

To assess the extent of RoCKseq modification, a fluorescent assay that tests the binding capacity of distinct fluorescent probes was implemented (Fig. 1c; see also Saikia et al.[33]). Using this assay, we verified that various modification rates can be easily obtained using the RoCKseq bead modification protocol (Fig. 1d). This is achieved using a mixture of the splint and an oligo that is complementary to the TSO sequence but lacks the capture sequence (TSO titration oligo, Supplementary Fig. 1c). In addition, this experiment shows that the bead modification does not alter bead integrity (including dT oligos; Fig. 1d) or size (Supplementary Fig. 1d–f). Furthermore, we successfully validated multiplexed modification with three splints (Fig. 1e). The importance and efficiency of the lambda exonuclease step was tested

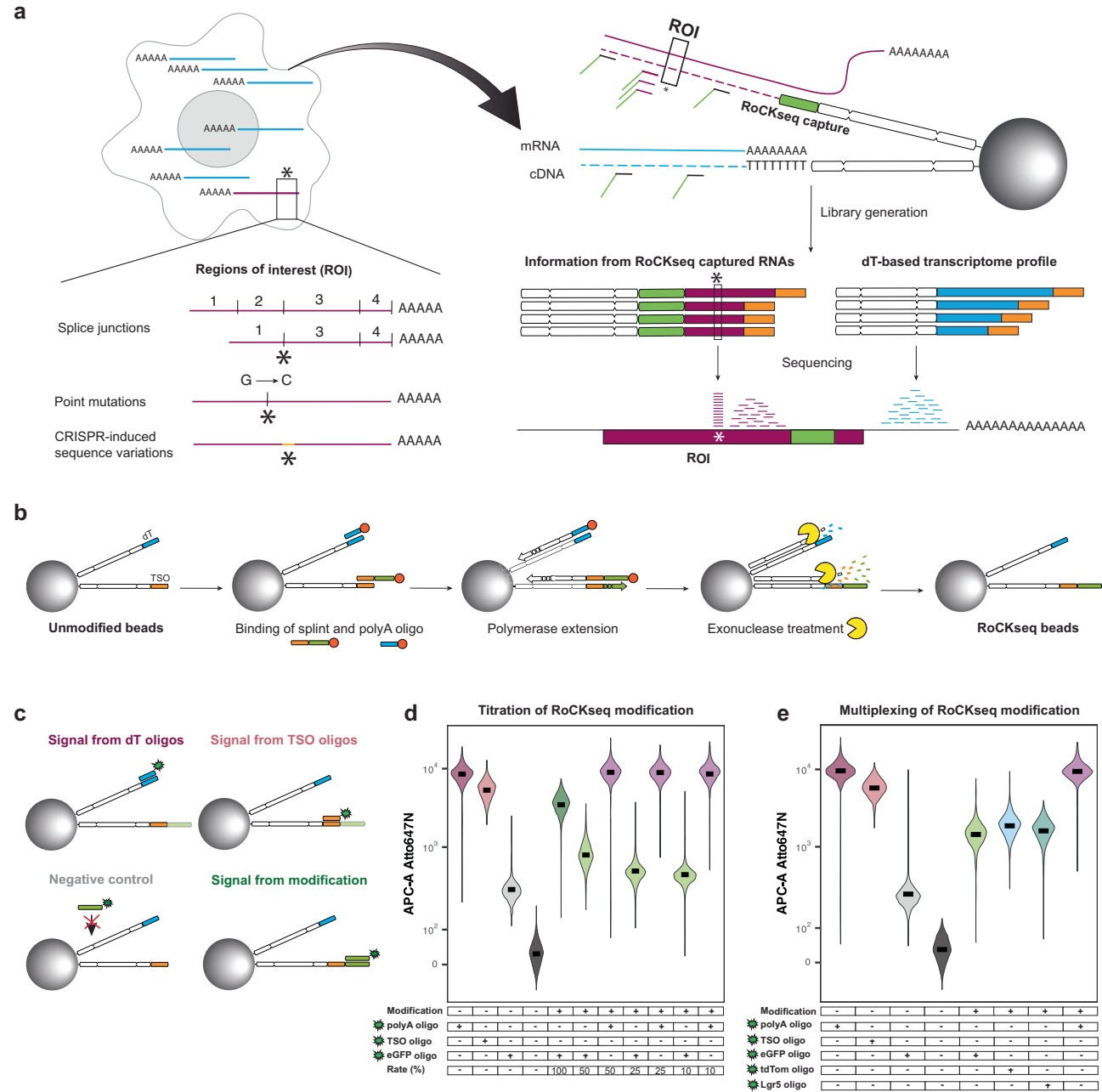

**Fig. 1 | RoCK and ROI concept and examples of RoCKseq bead modification.**
**a** Technique overview including BD Rhapsody beads modification (RoCKseq) and regions of interest enrichment (ROI) via primer addition. **b** RoCKseq bead modification. Red circles: 5' phosphate groups on oligonucleotides. **c** Fluorescent assay to assess bead modification and quality of oligos on the beads. Signal from dT oligos: polyA probe binding to the dT stretch on the beads; Signal from TSO oligos: probe complementary to the TSO; Negative control: probe complementary to the capture on unmodified beads; Signal from modification: probe complementary to the capture on the modified beads. **d**, **e** FACS quantification of RoCKseq bead modification. Titration of modification on RoCKseq beads ranging from 100% to 10% (**d**). Target: *eGFP* CDS. Modification of RoCKseq beads with multiple capture sequences in the same ratio (33% each) (**e**). Targets: *eGFP* CDS, *tdTomato* CDS, *Lgr5* CDS. To assess integrity of dT oligos on modified beads and to determine splint removal by lambda exonuclease, beads were tested using a polyA fluorescent oligo. For **d**, **e**, Y-axis: Atto647N fluorescent signal. The y axis has a biexponential transformation.

by comparing the standard treatment with a sample where either the entire step or the addition of the enzyme was omitted (Supplementary Fig. 3a). We observed that incubation with lambda exonuclease was necessary to fully restore the single strands on the beads (lower fluorescent signal for the other two conditions). In a next step, the effect of the protective polyA oligos used to prevent degradation of dT oligos by the T4 polymerase on the beads was tested. As shown in Supplementary Fig. 3b, both the dT and TSO oligos were degraded if they remained unprotected. The addition of the protective TSO oligo

during bead modification is thus important to keep into consideration when modification rates are lower than 100%.

To optimize the modification protocol, various other parameters were tested, including preincubation of the beads with the splint/polyA mix, prewarming of splints (Supplementary Fig. 3c, d) and purification level of oligos (Supplementary Fig. 3e). Importantly, we observe that RoCKseq modification is highly reproducible (Supplementary Fig. 3f) and modified beads remain stable over extended periods of time (at least 19 months; Supplementary Fig. 3g).

Taken together, these results show that standard BD Rhapsody barcoded beads can be reproducibly modified with custom capture sequences while maintaining bead integrity and with low variation among the pool of modified beads. Furthermore, distinct modifications (multiplexed capture sequences) can be easily combined and the rate of modification is scalable, producing custom RoCKseq beads that remain stable for months.

### Development of a simple protocol to direct reads to regions of interest

To direct reads to regions of interest, we developed ROIseq, in which a specific primer (or multiple primers for multiple ROIs) is (are) spiked into the pool of randomers during library generation (Supplementary Fig. 4a, b). Randomers are random primers of nine nucleotides to which an adapter is attached, and which are used to generate cDNA second strands in the BD Rhapsody platform. Importantly, the addition of randomers leads to the generation of random 5′ ends for the cDNAs (Supplementary Fig. 4a). By specifically designing primers targeting regions of interest (ROIs) on target RNAs, we can enrich for pre-defined 5′ ends of the corresponding cDNAs and thus specifically guide the reads obtained by HTS-based analysis to the ROIs in the target transcript (Supplementary Fig. 4b). Importantly, the standard randomers used for library generation are also included to profile the cell's transcriptome. To obtain information on both the transcriptome of a cell and the targeted capture library in the same experiment, a library generation protocol was developed (Supplementary Fig. 4b). The library entails four main changes to the standard BD Rhapsody library generation protocol (Supplementary Fig. 4a). First, a T primer (specific to TSO oligos on the BD Rhapsody beads; see Supplementary Fig. 1a, b) is added during second strand PCR amplification to retrieve information from the RoCKseq captured transcripts. Second, ROIseq primers are added to the pool of randomers to direct reads to regions of interest. Next, a custom indexing primer is used for the indexing of the RoCKseq capture library. This leads to the generation of two separately indexed libraries, one derived from the dT oligos (WTA library) and the other from TSO oligos (TSO library) that are mixed for HTS sequencing. Finally, a custom primer is added during HTS sequencing to retrieve information from TSO libraries.

### A custom, automated workflow to analyze targeted and untargeted data

We have designed an open-source Snakemake[45] workflow to process data from raw sequencing reads, leveraging the BD Rhapsody dual oligos present on the barcoded beads, with distinctive cell barcode structure differentiating the targeted (TSO) from untargeted (WTA) data (Supplementary Fig. 1b; see "Methods"). The workflow (Fig. 2a) generates a transcriptome index to match the experimental design (i.e., taking into account the cDNA read length). After indexing with STAR[46], FASTQ files are aligned and counted using STARsolo[47] while extracting valid cell barcodes and producing count tables for TSO and WTA readouts separately. We provide other running modes to deal with ad-hoc use cases, such as targeting repetitive sequences and hence including multimapping reads. Aside from producing count tables, our workflow generates basic scRNA-seq analysis reports, including quality control and cell clustering.

### Addition of T primer does not affect WTA information

To test RoCK and ROI, we wanted to confirm that the addition of the T primer does not affect the WTA readouts. Additionally, since this primer is needed to obtain information from TSO oligos, we aimed to explore the generation of a TSO oligo-based library (TSO library) given by the capture of transcripts on these oligos. For this assessment, we chose two clonal cell lines, each expressing distinct fluorescent proteins. We generated a 1:1 mix of clonal (human) HEK293-T cells

expressing tdTomato and clonal (murine) L-cells expressing eGFP, both of which were produced by lentiviral transduction (Supplementary Fig. 5a). We generated libraries using unmodified beads and a standard BD Rhapsody protocol, either with (unmod_T, WTA and TSO libraries) or without (unmod, WTA library) the addition of the T primer (Fig. 2b, Supplementary Table 1 and Supplementary Data 1). A first evaluation of the libraries before indexing did not reveal any noticeable difference between the two conditions (Supplementary Fig. 5b). We then reasoned that the addition of the T primer would have generated cDNAs from transcripts that have been captured by the TSO sequence (TSO library). We therefore indexed the standard dT libraries (with and without T primer) as well as the TSO library generated from the putative TSO captured RNAs. As before, the two final dT libraries had very similar characteristics (Supplementary Fig. 5c). As presumed, also a TSO library was generated with a similar trace as the dT libraries. This indicates that the addition of the T primer allows retrieval of information that derives from transcripts captured via the TSO. After sequencing and single-cell read mapping and counting, we first checked whether the information from the two WTA libraries was similar in terms of number of genes (Supplementary Fig. 5d), number of UMIs (Fig. 2c and Supplementary Fig. 5e) and percent of mitochondrial content (Fig. 2c and Supplementary Fig. 5f) detected per cell. This was the case for both human and mouse cells. The two cell types could be clearly distinguished based on the WTA libraries (Fig. 2d). To determine if the T primer addition affects the WTA readout, we pairwise compared the per-gene counts obtained with and without addition of the T primer. The transcriptomes of the unmod and unmod_T conditions were similar (Fig. 2e, f, Pearson correlation 0.976 for mouse, 0.974 for human), with most genes showing minimal changes in expression (log2FC ≈ 0) and only a small subset being up- or downrepresented (Supplementary Fig. 5g, h), indicating that the T primer does not hamper library generation nor significantly alter the WTA signal derived from dT oligos.

### The targeting strategy does not bias the WTA information

We next performed an additional RoCK and ROI experiment using the same cell lines (1:1 mix of eGFP or tdTomato expressing cells). The aim of the experiment was to compare the *eGFP* and *tdTomato* detection sensitivity with and without targeted capture (RoCKseq) and ROIseq-based priming. To capture both transcripts, we selected a 25 bp stretch at the 3′ end of the CDSs that is shared between *eGFP* and *tdTomato* (Supplementary Fig. 5i), allowing a single configuration of RoCKseq beads (Supplementary Fig. 6a). A single ROIseq primer for *eGFP* and two ROIseq primers for *tdTomato* were used. As both transcripts share the 5′ and 3′ untranslated region (UTR) sequences, they can only be distinguished by reads from their respective CDSs (Supplementary Fig. 5a).

To assess the individual effects of RoCKseq capture and ROIseq primers, we tested four experimental conditions (Fig. 3a, Supplementary Table 1 and Supplementary Data 1): the standard BD Rhapsody protocol using (1) unmodified beads (unmod) or (2) the unmodified beads with ROIseq and T primers (unmod_roi); and RoCKseq beads (3) without (rock) and (4) with the addition of ROIseq primers (rockroi). Initial quality control on libraries before and after indexing (Supplementary Fig. 6b, c, respectively) showed that global properties of the WTA and TSO libraries were similar for all conditions.

The four conditions had similar WTA transcriptomes in terms of number of genes, number of transcripts and percent mitochondrial content (Fig. 3b and Supplementary Fig. 6d–f). In addition, the information in the WTA libraries was sufficient to distinguish between mouse and human cells in all conditions (Supplementary Fig. 6g). The similarity of the average transcriptional profiles across conditions was apparent when comparing the information obtained from the WTAs for mouse (Fig. 3c, Pearson correlations between 0.982 and 0.989) and

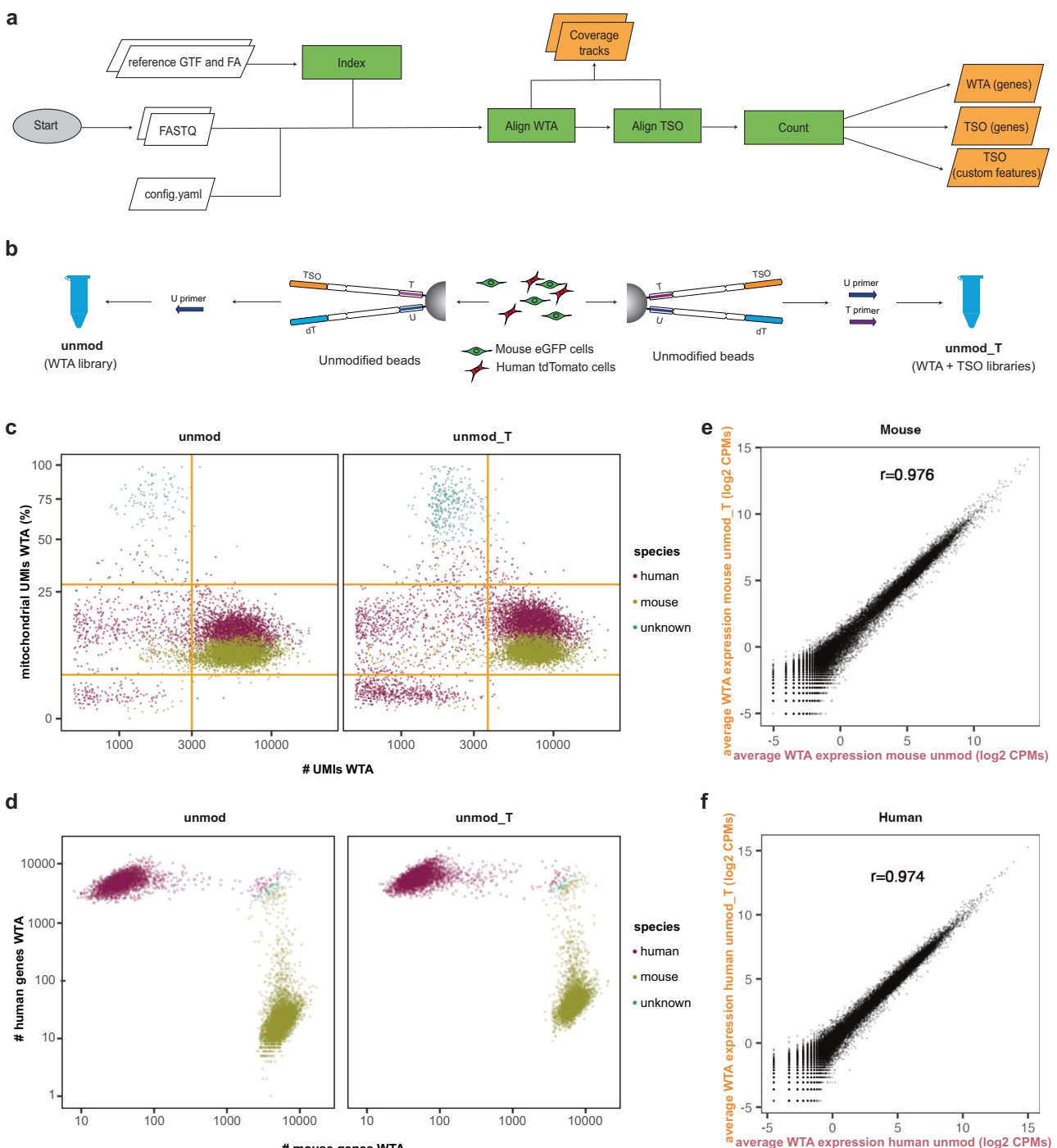

**Fig. 2 | Analysis workflow and testing of the effect of addition of T primer on the WTA of unmodified beads. a** RoCK and ROI data analysis pipeline (see "Methods"). **b** Experimental setup (mixing scRNA-seq experiment) including unmod (Universal primer; U primer) and unmod_T (U and T primers) conditions. **c** QC of WTA data depicting filtering thresholds (orange). **d** Barnyard plot depicting cell species assignment using WTA data. **e** Correlation of WTA readouts in unmod_T vs. unmod conditions, mouse cells only. **f** Same as (**e**) for human cells.

human (Fig. 3d, Pearson correlations between 0.983 and 0.986) samples, with low to no changes in gene expression for most genes (log2FC ≈ 0) across gene expression levels (Supplementary Fig. 7a–l).

We next focused on the CDS detection for the *eGFP* and *tdTomato* transcripts. Compared to the unmod condition, unmod_roi and particularly rock and rockroi showed an increase in the number of cells with at least one detected UMI in the respective *eGFP* and *tdTomato* CDS (Fig. 4a). The increase was particularly apparent in the rock and rockroi conditions, indicating that RoCKseq capture strongly aids with the

detection of the CDS. This effect can be seen when looking at the coverage along the *eGFP* and *tdTomato* transcripts in mouse and human cells, respectively (Fig. 4b, c). Compared to rock, in the rockroi condition a single prominent peak of reads in the *eGFP* transcript, precisely where the ROIseq primer had been positioned, is visible. Similarly, two distinctive coverage signal peaks can be seen in the rockroi condition for the *tdTomato* transcript. Since *tdTomato* was generated by fusing two copies of the dTomato gene to create a tandem dimer[48], we retained multimapping alignments, hence reporting

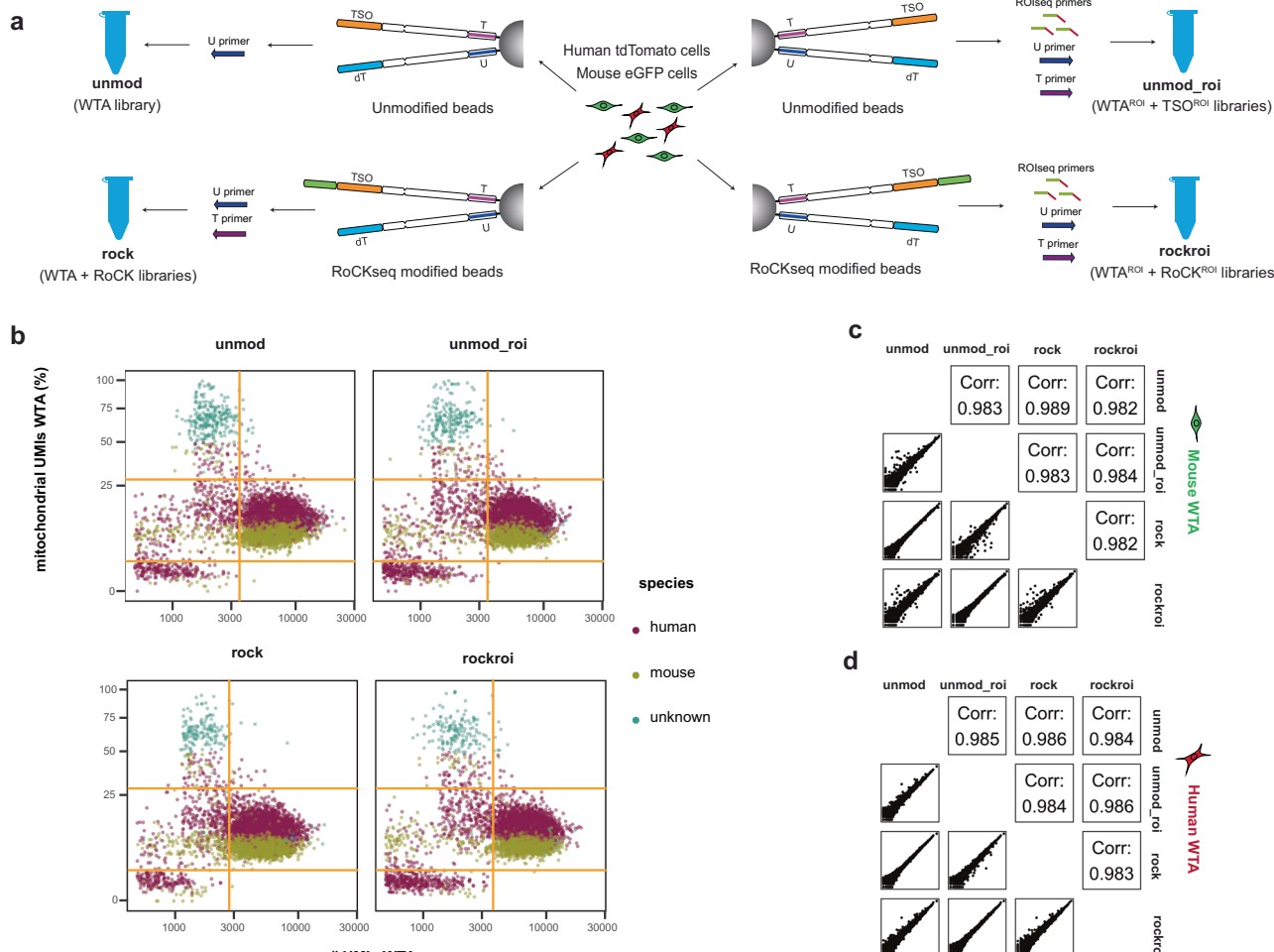

**Fig. 3 | Analysis of RoCK and ROI WTA data from mixing experiment.**
**a** Experimental setup of the mixing experiment with extended conditions, including unmodified beads (unmod, unmod_roi) and modified beads (rock, rockroi). Primers: U (all conditions); T: unmod_roi, rock, rockroi; ROIseq: unmod_roi and rockroi conditions. **b** QC of WTA data depicting filtering thresholds (orange). **c** Pairwise correlation of WTA data across conditions (mouse cells only). **d** Same as (**c**) for human cells. Corr: correlation.

alignments twice. Of note, when comparing the unmod_roi with the rockroi condition, the need for RoCKseq capture when targeting sequences of interest becomes apparent, as only a small peak of reads is visible in the unmod_roi condition. This can be explained by the distance of the CDS to the polyA tail being >1.5 kb in all cases. The WTA coverage remained very similar across conditions (Supplementary Fig. 8a, b), while the increase in sequencing coverage of the *eGFP* and *tdTomato* CDSs is largely driven by TSO reads (Fig. 4b, c). This effect is also apparent when comparing the numbers of counts per cell derived from WTA vs. TSO (Fig. 4d).

When comparing the number of counts for the *eGFP* ROI in the unmodified condition to the detection rate of the same region in the other conditions, we found a 2423 fold enrichment for the rockroi condition, while these numbers were 205.7 for rock and 100.7 for unmod_roi (Fig. 4e). For the ROI 1/3 (Fig. 4f) and ROI 2/4 (Fig. 4g) regions of *tdTomato*, these numbers were 2893.17 and 1962.9 for the rockroi condition, 326.0 and 455.86 for rock, 36.6 and 21.48 for unmod_roi, respectively. These results demonstrate that RoCK and ROI targeted priming and capture act synergistically and substantially enhances the retrieval of ROI information on the targeted transcripts.

We next looked at the percent of cells with detectable *eGFP* and *tdTomato* expression. Due to the *eGFP* and *tdTomato* sequence similarity we consider as positive cells those with at least one alignment to the targeted CDSs, unique or not. Compared to the unmod condition, where the CDSs of *eGFP* and *tdTomato* were detected in 4.80% and 8.17% of cells, respectively, RoCKseq capture increased the detection to 98.83% *eGFP*-positive L-cells and 98.18% *tdTomato*-positive HEK293-T cells (Fig. 4h). The addition of the ROIseq primer (unmod_roi) increased the detection of the *eGFP* CDS and *tdTomato* CDS to 23.25% and to 32.67%, respectively, while in rockroi an even higher proportion of positive cells was observed (*eGFP* CDS: 99.37%; *tdTomato* CDS: 99.56%). The detection of *eGFP* and *tdTomato* was highly specific, with very low false positives for the RoCKseq and ROIseq regions (for mouse cells: Supplementary Fig. 8c, for human cells: Supplementary Fig. 8d).

To evaluate the sensitivity of RoCK and ROI, we wanted to understand how the number of UMIs relates to the number of targeted RNAs present in a cell. Previous reports have shown that only 5–20% of the transcriptome of a cell is recovered in scRNA-seq experiments[6–8,49,50]. To determine the number of *eGFP* transcripts expressed in a single cell, we visualized single transcripts by RNAscope on the same clonal L-cell line (Supplementary Fig. 8e–g). As a negative control, we used untransfected L-cells (wt) or a clonal L-cell line expressing tdTomato. For *eGFP* transcripts, we quantified 30–233 spots (average 118, median 118.5) for the first replicate and 58–336 spots (average 131, median 126) for the second replicate, both varying according to cell size (Supplementary Fig. 8h, i: RNAscope

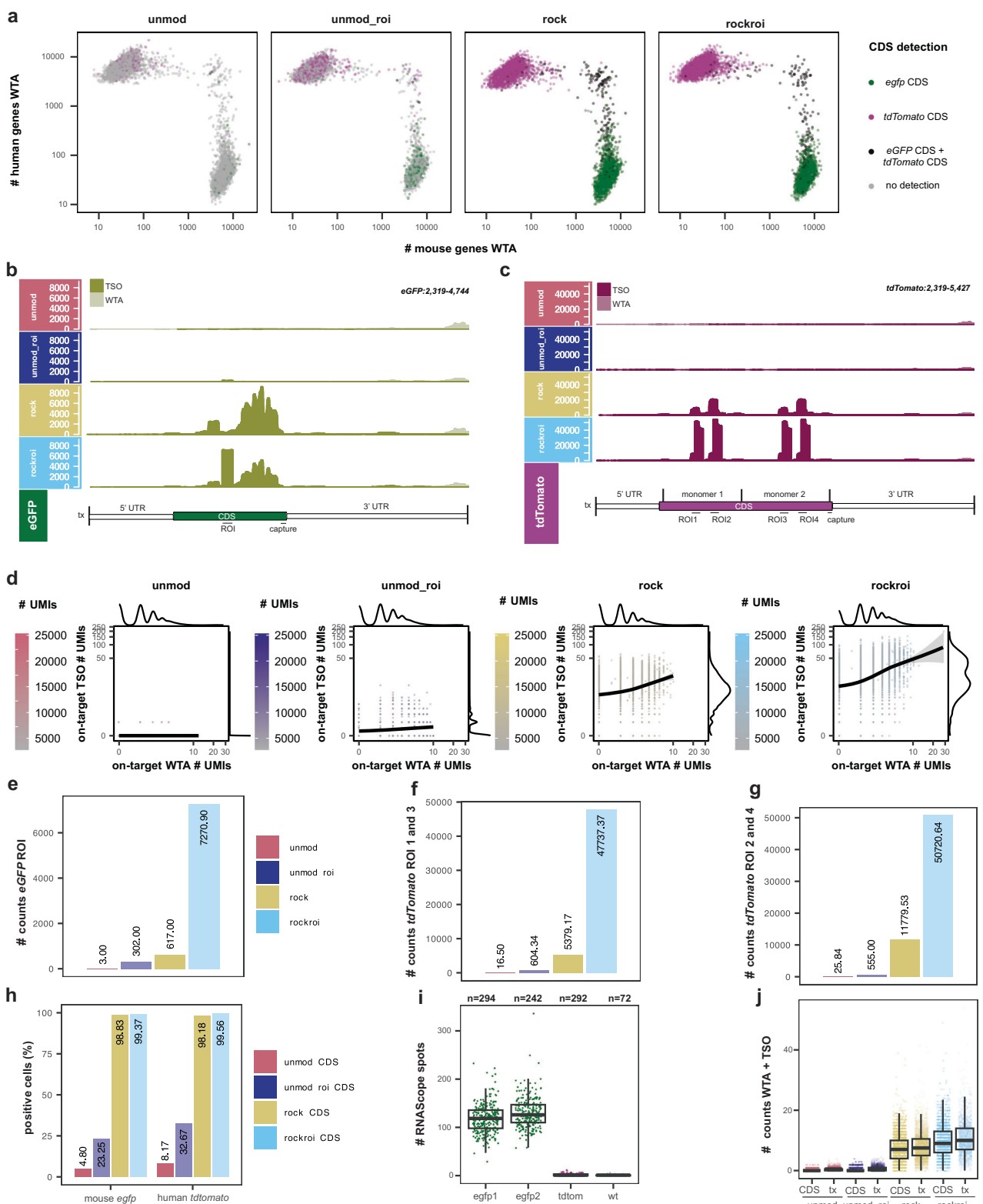

spots normalized by area). On the other hand, an average of 0.44 counts per cell were measured in the scRNA-seq experiment for the full *eGFP* transcript (CDS plus UTRs) in the unmod condition, 7.95 counts for the rock condition and 10.68 counts per cell for the rockroi condition (Fig. 4i, j). This indicates that we detect 0.35% of *eGFP* transcripts per cell for the unmod condition and 8.58% for the rockroi condition, thus reaching similar detection rates for transcripts as determined for other methods in previous reports[6–8,24,49,50].

The detection limit of RoCK and ROI for very lowly expressed transcripts was investigated in a targeted experiment performed on HeLa cells and the results are summarized in Supplementary Information (Supplementary Note 1 and Supplementary Figs. 9–12).

Altogether, RoCKseq leads to a drastic increase in the detection of transcripts of interest and in combination with ROIseq primers, RoCK and ROI enriches reads in regions of interest. This targeted information is recorded together with the WTA of cells.

**Fig. 4 | Analysis of RoCK and ROI target enrichment data and quantification of *eGFP* mRNAs. a** Barnyard plot colored by detection of *eGFP* and *tdTomato* CDS in WTA and TSO data. **b** Sequencing coverage and depth along *eGFP* in mouse cells for TSO (olive green) and WTA (off white). **c** Sequencing coverage and depth along *tdTomato* in human cells for TSO (red purple) and WTA (light mauve). Bars at the bottom of **c** and **d** indicate regions in the *eGFP* and *tdTomato* transcript and CDS, also used for counting. The CDS is indicated in green or magenta, respectively, while the UTRs are indicated in white. **d** Detection of *eGFP* and *tdTomato* in TSO vs. WTA data, per cell. Number of counts for ROI^eGFP (**e**), sum of ROI 1 and 3 (**f**) and sum of ROI 2 and 4 (**g**). **h** Percent of cells with detectable *eGFP* CDS (mouse cells) or *tdTomato* CDS (human cells) in TSO plus WTA data, per condition. **i** Number of

*eGFP* mRNAs in mouse cells detected by RNAscope. egfp1 and egfp2: biological replicates. Negative controls: L-cells expressing *tdTomato* and wt L-cells (untransduced). Number of analyzed cells: eGFP1 $n = 294$, egfp2 $n = 242$, tdTomato $n = 292$, wt $n = 72$. Boxplots show the median (center line), the first and third quartiles (bounds of the box; 25th and 75th percentiles), the lower and upper whiskers (1.5 × IQR from the box). Source data are provided as a Source Data file. **j** Number of counts from combining WTA and TSO data for the *eGFP* CDS and transcript (tx). Boxplots show the median (center line), the first and third quartiles (bounds of the box; 25th and 75th percentiles), the lower and upper whiskers (1.5 × IQR from the box).

## Characterization of RoCKseq capture and ROIseq targets

We next looked specifically into the TSO modality for reads that did not map to our targeted regions. For the scRNA-seq experiment described in Fig. 3a, the percentage of (on-target) *eGFP*- or *tdTomato*-specific TSO alignments was 0.38%, 0.20% and 0.008% for the rockroi, rock and unmod_roi conditions, respectively (Supplementary Fig. 13a), indicating low specificity. This was also apparent when looking at the number of genes and UMIs detected in the TSO data in all samples in which the T primer was added, independent of bead modification (Fig. 5a, b) and was also true for the scRNA-seq experiment described in Fig. 2b (Supplementary Fig. 13b, c). The percentage of intergenic information was slightly higher in WTA compared to TSO libraries (Supplementary Fig. 13d). Additionally, the TSO information in genes showed a higher percentage of non-protein coding genes compared to the WTA libraries (Supplementary Fig. 13e), including also non-polyadenylated types, which may be explained by internal capture of transcripts. In fact, when looking at the TSO coverage across gene bodies, it was not biased towards the transcript 3′ end (as is the WTA readout; Fig. 5c). This was true for both scRNA-seq experiments and thus independent of the bead modification (Supplementary Fig. 13f). Compared to the WTA readouts, the unmod_T TSO modality showed a higher percentage of mitochondrial transcripts per cell (Fig. 5d). This was also apparent when looking at the coverage across the detected mitochondrial transcripts (Fig. 5e, f).

TSO libraries showed a lower number of genes (Supplementary Fig. 6d vs. Fig. 5a), UMIs (Supplementary Fig. 6e vs. Fig. 5b) and reads with canonical cell barcodes (Supplementary Fig. 13g) compared to the WTA modality, although the libraries were mixed at a 1:1 concentration. The TSO libraries also had a lower number of alignments compared to the WTA libraries (Supplementary Fig. 13h). To look into this, we tracked the reads and alignments across the conditions of the two mixing experiments at different relevant steps for the data analysis (Supplementary Fig. 14a–l). We noticed most WTA aligned reads belonged to high-quality (retained) cell barcodes regardless of the bead modification, whereas most TSO reads did not. This difference is also reflected when looking at the total number of UMIs in the TSO and WTA count tables. The discrepancy in the amount of information deriving from WTA and TSO modalities is thus occurring already at the sequencing step and is further affected by downstream processing steps.

Although we observed a difference in information content between WTA and TSO and the percentage of *eGFP*- or *tdTomato*-specific TSO alignments is low, the number of on-target UMIs was higher in TSO vs. WTA data in all conditions in which the T primer was added. This was especially true for the rock and rockroi conditions, where an 80-fold and 94-fold increase in on-target (unique or not) alignments in rock and rockroi was observed, respectively, in the TSO data compared to WTA (Supplementary Fig. 14j, l).

Similar to the RoCKseq capture, we asked if the ROIseq primers bind in transcripts other than the targeted *eGFP* and *tdTomato*. We observed that the ROIseq primers were binding to off-target RNAs, leading to ROIseq-specific peaks on both WTA and TSO modalities (Fig. 5g–i). The low-stringency annealing conditions used during

second strand synthesis give rise to mispriming of the ROI primer, resulting in chimeric (artifact) cDNA reads (referred to as artificially primed products; APPs) for both TSO off-targets and WTA. These APP cDNA reads start with the respective ROI sequence. However, we found that the WTA in modified beads is highly similar to that of unmodified beads (Fig. 3c, d), indicating that neither RoCKseq nor ROIseq had a major impact on the overall untargeted transcriptome.

## Comparison of RoCK and ROI to other targeted scRNA-seq methods

To put RoCK and ROI into context with other targeted scRNA-seq methods, we compared parameters including level of targeting, detection of standard transcriptome, dry-lab workflow, cell throughput, polyA-independent detection of targets and sequencing cost (Table 1). We also considered parameters such as protocol time, maximal number of cells which can potentially be analyzed per experiment, cost per cell and sensitivity of targeting while keeping into account read depth per cell (Supplementary Tables 2–9). Out of the methods we compared, RoCK and ROI is the only one combining targeting at both RNA level and first strand without additional cDNA level amplification in combination with the unchanged information on the WTA. Importantly, RoCK and ROI only introduces a short increase in processing time and is comparable to other targeted methods in terms of cost. On the other hand, our method leads to a strong enrichment in detection of targets of interest compared to the unmodified sample. These features position RoCK and ROI as a versatile and efficient approach, uniquely combining robust target enrichment with comprehensive transcriptome information while maintaining cost and processing efficiency.

## RoCK and ROI enables the detection of *Pdgfrα* splice junctions

After validation of RoCK and ROI in cell lines, we chose the murine colon as a complex biological system with multiple transcriptionally-distinct cell types (Fig. 6a, Supplementary Table 1 and Supplementary Data 1). First, we wanted to test if the WTA modality from a RoCK and ROI experiment can identify and annotate the same cell types as that from an unmodified bead experiment. Second, we wanted to quantify splice junctions of a targeted transcript. We chose a mouse strain where the H2B-eGFP fusion protein reporter construct was knocked into one of the *Pdgfrα* alleles[51] (Supplementary Fig. 15a, b), where *Pdgfrα* is a marker for mesenchymal cells. Of note, the *Pdgfrα* gene (and the *eGFP* reporter) is expressed at different levels in crypt top and crypt bottom fibroblasts[52]. Several protein-coding transcripts are encoded by the wildtype *Pdgfrα*, including short transcripts with 16 exons and long transcripts with seven additional exons. For RoCK and ROI, the beads were modified with 1:1:1 ratio for three capture sequences: *eGFP*:*Pdgfrα*-targeting-exon-7:*Pdgfrα*-targeting-exon-17 (Supplementary Fig. 15c). In addition, eight ROIseq primers were spiked in during library generation, one for *eGFP* detection (ROI^eGFP) and seven for *Pdgfrα* (ROI^Pα) to probe splice junctions nearer to the transcript's 5′-prime end, where usually no information is retrieved in scRNA-seq experiments (Supplementary Fig. 15d, e).

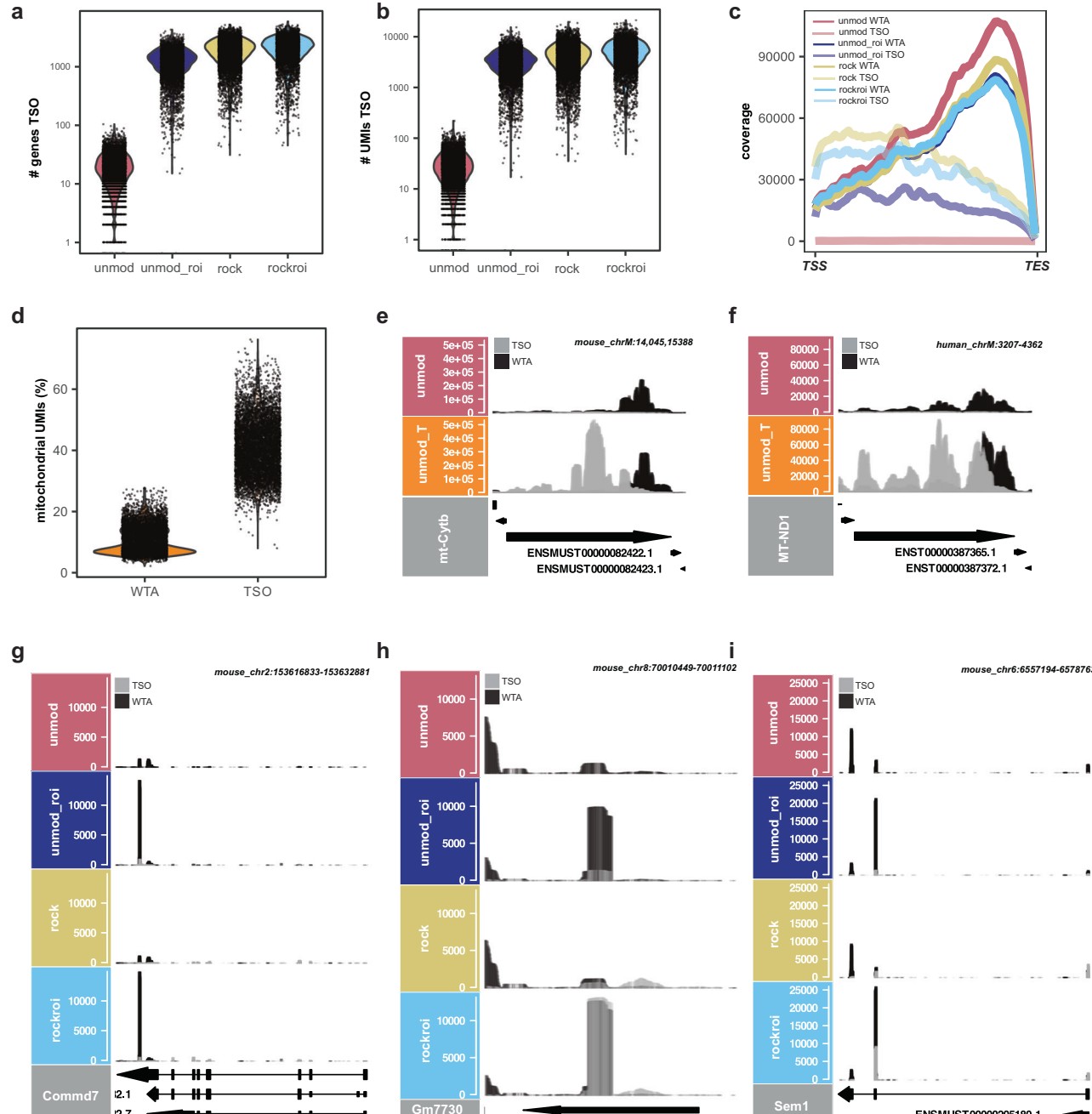

**Fig. 5 | Characterization of RoCK and ROI TSO data and example of ROIseq peaks. a** Number of genes detected in the TSO data. **b** Number of UMIs detected in the TSO data. **c** Aggregated sequencing coverage along detected transcripts for TSO and WTA data; TSS transcription start site, TES transcription end site. **d** Mitochondrial content in WTA and TSO data. Sequencing coverage for TSO (gray) and WTA (black) along *mt-Cytb* (**e**), *MT-ND1* (**f**), *Commd7* (**g**), *Gm7730* (**h**) and *Sem1* (**i**). Data in **a**–**c** and **g**–**i** refers to experiment described in Fig. 3a, data in **d**–**f** refers to experiment described in Fig. 2b.

We performed the experiment with a 1:1 mixture of sorted eGFP-positive colonic fibroblasts and Epcam-positive epithelial cells (Supplementary Fig. 16a, b), using either unmodified beads (unmod) or RoCK and ROI (rockroi). To simplify the comparison of the WTAs, we combined and sequenced the WTA profiles of the unmod and rockroi libraries in a full cartridge (unimodal condition, WTA and WTA^ROI libraries). In a second cartridge, we sequenced the WTA and TSO libraries of the rockroi condition (multimodal condition, WTA^ROI and TSO^ROI). As the custom sequencing primer is used to obtain information on the TSO libraries, it was not added to the cartridge in which exclusively WTA profiles were recorded.

As in previous experiments, the WTA sensitivity for the unmod and rockroi samples looked similar in terms of number of genes, number of UMIs and mitochondrial content (Supplementary Fig. 17a–c). We then manually annotated epithelial and fibroblast clusters using known markers (see Brügger et al.[52]; Supplementary Fig. 17d and Supplementary Table 10). All cell types detected in unmod were also detected in rockroi (Fig. 6b), including rare cell types, such as Tuft and enteroendocrine cells. The detected mitochondrial content (Fig. 6c) and genes (Fig. 6d) across clusters were similar between the unmodified and rockroi conditions.

The ROI^Pα primers added during library generation yielded reads spanning splice junctions in the *Pdgfrα* transcript (Fig. 6e, f). As

**Table 1 | Overview of targeted scRNA-seq techniques applicable to custom targets**

| Technique name | RNA level targeting | First strand level targeting | No cDNA level amplification | Standard transcriptome | Dry-lab workflow | High cell throughput | Detection of non-polyadenylated targets | Sequencing cost |
|---|---|---|---|---|---|---|---|---|
| RoCK and ROI | + | + | + | + | + | + | + | $$ |
| DART-seq[33] | + | – | + | + | + | + | + | $ |
| HyPR-seq[24] | + | – | + | – | + | + | + | $ |
| TARGET-seq[58] | + | – | – | + | + | – | – | $$ |
| BART-seq[59] | – | + | + | – | + | – | + | $ |
| TAPseq[38] | – | + | – | + | + | + | – | $$ |
| GoT-seq[20] | – | + | – | + | + | + | – | $$$ |
| GoT-Splice[40] | – | + | – | + | + | + | – | $$$ |
| Chigene[60] | – | + | – | + | – | + | – | $$ |
| BD Targeted Amplification[28] | – | + | – | – | – | + | – | $$ |
| Constellation-seq[23] | – | + | – | – | – | + | – | $$ |
| RAGE-seq[30] | – | – | – | + | + | + | – | $$$ |
| Van Horebeek et al.[26] | – | – | – | + | – | + | – | $ |
| scTaILoR-seq[36] | – | – | – | + | – | + | – | $$$ |
| scTLR-seq[39] | – | – | – | + | – | + | – | $$$ |
| scCapture-seq[29] | – | – | – | + | – | – | – | $ |

RNA level targeting, First strand level targeting and No cDNA level amplification refer to the step in which the targeting occurs. As targeting at the cDNA level introduces additional biases and is performed at a later step compared to the first two levels in columns 2 and 3, the header was switched to no cDNA level amplification to highlight that this is an advantage. Standard transcriptome: ability of the method to retrieve transcripts other than targets. Dry-lab workflow: whether a tailored data analysis workflow is provided. High cell throughput: ability to profile a high number of cells. Detection of non-polyadenylated targets: whether non-polyadenylated RNAs such as lnc-RNAs and miRNAs can be detected. Sequencing cost: $: a single short read library is needed; $$: multiple short read libraries need to be sequenced to obtain the targeted and untargeted data; $$$: long read sequencing is required. Methods are sorted by total number of desirable properties, namely: RNA level or first strand level targeting, lack of cDNA level amplification/targeting, profiling the standard transcriptome, availability of a dry-lab workflow, high cell throughput and low sequencing cost.

expected, these reads were detected exclusively in fibroblast clusters. Additionally, in most of the junctions targeted by ROIseq, the percent of positive cells was higher in crypt top compared to crypt bottom cells, which is consistent with previous findings showing that crypt top fibroblasts have a higher *Pdgfrα* (and *eGFP*) expression[52]. In contrast, reads in the regions targeted by ROIseq primers were completely absent in the unmodified sample (Supplementary Fig. 17e–g) but clearly yielded reads spanning the targeted splice junctions in rockroi (Supplementary Fig. 17g). In addition to *Pdgfrα*, we also detected *eGFP* (Supplementary Fig. 18a–c), again with exclusive expression in fibroblasts.

We next compared the cell types detected via the WTAs of the unmod and rockroi conditions to determine if adding a set of ROIseq primers affected the ability to distinguish distinct subpopulations in an scRNA-seq experiment. The cell types detected in the unmod and rockroi samples were highly concordant (Fig. 6g, Pearson correlation between 0.94 and 0.97), indicating that the addition of multiple ROIseq primers during library generation does not significantly impact the WTA profiles.

We then shifted our focus to the WTA profiles of the unimodal vs. multimodal rockroi conditions. Since the same library was sequenced twice, this gives a baseline of technical variation; the two WTA readouts were highly correlated (Pearson correlation 0.987, Supplementary Fig. 18d, e).

To discriminate between *Pdgfrα* long and short transcripts, we looked into the discriminant splicing region between exons 16 and 17 (Supplementary Fig. 15b, d, e). First, our RoCKseq capture in exon 17 is specific to the long transcripts. Second, the junction where discriminant splicing occurs is also targeted by the roi_16 primer; reads spanning this exon junction are specific to the *Pdgfrα* long transcripts. The short isoforms on the other hand can be detected by reads mapping to the 3' UTR of the short *Pdgfrα* isoform, which are present in

both rockroi and unmodified samples in crypt bottom and top cells (Supplementary Fig. 18f, g).

Taken together, RoCK and ROI is able to direct reads to specific regions of interest such as splice junctions. By capturing *Pdgfrα* close to a junction of interest and adding a primer spanning this region, RoCK and ROI can also detect and distinguish between splice variants. Furthermore, the concomitant WTA data allow to cluster and annotate cells based on their standard transcriptome, hence allowing to dissect splicing outcomes for specific cell types.

### *BCR::ABL1* fusion detection in leukemic cell lines and patient samples

The *BCR::ABL1* fusion transcript is one of the best characterized causative events leading to chronic myeloid and acute lymphoblastic leukemia[53]. Typical *BCR::ABL1* fusion breakpoints can be distinguished between major and minor breakpoints depending on the fusion junction in the *BCR* gene (exon 1 for minor, e1a2, and exon 13 or 14 for major, e13a2 and e14a2 respectively). The position of the *ABL1* fusion breakpoint is relatively conserved across the major and the minor fusions and leads in nearly all cases to the fusion of exon 2[54]. In this case, the fusion junction is far away from the 3' end of the transcript at the cDNA level (3'313 bp) and *BCR::ABL1* fusion transcripts are notoriously difficult to detect in scRNA-seq experiments[55]. An additional challenge in the analysis of *BCR::ABL1* fusion transcripts in scRNA-seq experiments is the fact that cells express both a fused (*BCR::ABL1*) and non-fused (wild-type) transcripts of *BCR* and *ABL1*. The only way to distinguish between the fused and non-fused transcripts is thus to direct reads across the fusion point. To achieve this goal, we designed a single RoCKseq capture sequence in the *ABL1* transcript close to the fusion junction (Fig. 7a). Furthermore, to distinguish between the minor and major breakpoints, we designed two ROIseq primers: one in *BCR* exon 1 and one in *BCR* exon 13. Since exon 14 of *BCR* is short

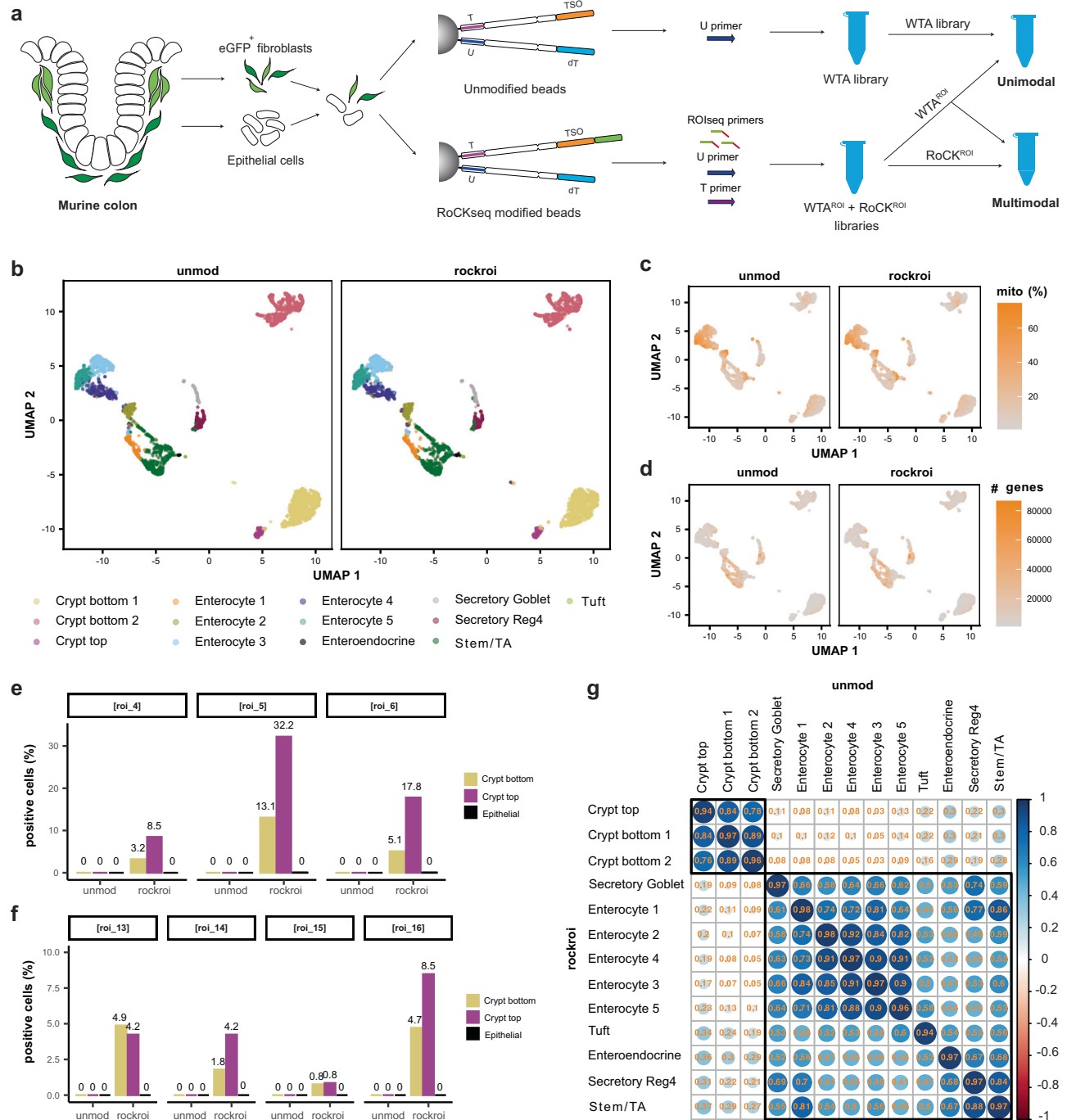

**Fig. 6 | Detection of *Pdgfrα* splice junctions in murine colon cells.**
**a** Experimental set up of the *Pdgfrα* experiment including eGFP+ fibroblasts and EpCAM+ epithelial cells from murine colon. Conditions: unmod and rockroi (unimodal and multimodal); Primers: U (all conditions); T and ROIseq (rockroi). **b** UMAP embedding on unimodal WTA data split by bead modification and colored by cell type (unsupervised clustering). Same UMAP colored by mitochondrial content (**c**) and by number of detected genes in WTA (**d**). **e**, **f** TSO detection rate of ROIseq-targeted splice junctions. **g** Pairwise Pearson correlation of gene (WTA) expression readouts across cell types in unmod (horizontal) vs. rockroi conditions (vertical).

(75 bp), the ROIseq primer in exon 13 allows detection of both major fusions using a long-enough read length (180 bp for this experiment).

To test our RoCK and ROI targeted enrichment strategy, we used a mix of three patient-derived leukemia cell lines containing either a minor fusion (SUP-B15, e1a2), major fusion (K562, e14a2) or no *BCR::ABL1* fusion (Loucy) (Fig. 7b, Supplementary Table 1 and Supplementary Data 1). We first manually annotated the cell line clusters based on the dT captured data (Fig. 7c and Supplementary Fig. 19a) and subsequently re-aligned the reads to ChimerDB 4.0[56], a database of

fusion genes including *BCR::ABL1* fusions (see "Methods"). We applied a custom alignment strategy to separate reads congruent with a *BCR::ABL1* gene fusion origin from ROI[BCR]-containing artificially primed products (APPs), technical artifacts due to ROI[BCR] mispriming under permissive conditions; see "Methods" section and Supplementary Fig. 20. With this strategy, we were able to distinguish between the minor and major fusions (Fig. 7d, e). The e14a2 fusion was detected in 90.2% of K562 cells while the e1a2 breakpoint was detected in 18% of SUP-B15 cells (Fig. 7f). The detection was highly specific (e1a2 in Loucy

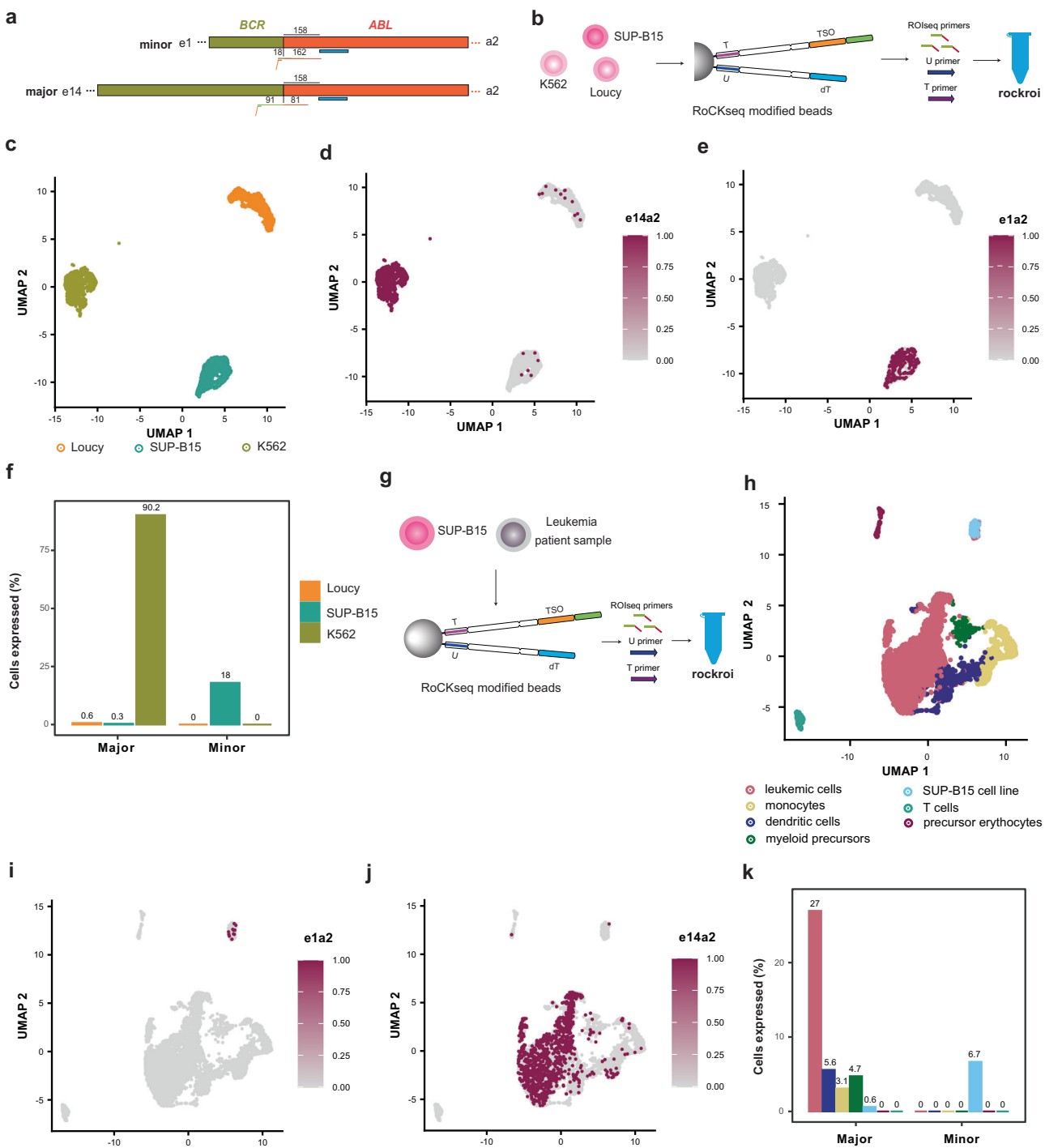

**Fig. 7 | Detection of the *BCR::ABL1* fusion transcript in leukemia cells.**
**a** Difference between major (e13a2 and e14a2) and minor fusions (e1a2) and RoCK and ROI targeting set up. **b** Experimental set up for detection of *BCR::ABL1* in cell lines. A mix of three patient-derived leukemia cell lines were used for sequencing (K562 with e14a2, SUP-B15 with e1a2 and Loucy with no breakpoint). Conditions: rockroi; Primers: U primer, T primer and ROIseq primers. **c** UMAP embedding on cell lines (unsupervised clustering). Same UMAP embedding colored by the e14a2 major breakpoint (**d**) and minor breakpoint (**e**). **f** Detection rate for minor and major breakpoints in the three cell lines (TSO data). **g** Experimental set up for detection of *BCR::ABL1* fusion transcript in the patient sample with an e14a2 breakpoint. The cells from the patient sample were mixed with a small amount of SUP-B15 cells. **h** UMAP embedding on patient and cell line data (unsupervised clustering). Same UMAP embedding colored by the e1a2 minor breakpoint (**i**) and major breakpoint (**j**). Positive cells are plotted on top. **k** Detection rate for minor and major breakpoints in patient sample (TSO data). Annotations for the cell types in the barplots are in (**h**).

cells 0% and e14a2 in Loucy cells 0.6%). Our detection rate for the fusion in K562 cells is in line with previous methods targeting the *BCR::ABL1* transcript[55].

We next applied our targeting and data analysis strategy to a *BCR::ABL1*-positive B-cell precursor acute lymphoblastic leukemia (BCP-ALL) patient sample at first diagnosis (Fig. 7g, Supplementary Fig. 19b, Supplementary Table 1 and Supplementary Data 1). Prior to the scRNA-seq experiment, routine diagnostic analysis of the patient sample had established an e14a2 major breakpoint. To test our ability to distinguish between the minor and major fusion in this context, we

spiked in a low amount of SUP-B15 cells expressing the *BCR::ABL1* e1a2 minor fusion. After clustering and manual annotation, we were able to distinguish between several clusters of immune cells (Fig. 7h and Supplementary Fig. 19c) and looked for e14a2 and e1a2 fusion points. As before, the minor breakpoint could be detected in the SUP-B15 cluster (Fig. 7i); additionally, the e14a2 fusion was observed in 27% of the patient's leukemic cells (Fig. 7j, k). Interestingly, the major fusion was detected in both lymphoid and myeloid lineages, extending recent results obtained with FISH on FACS-sorted hematopoietic cell populations indicating that the *BCR::ABL1* fusion is not restricted to the lymphoid lineage but also involves myeloid precursors[57]. Through our RoCK and ROI targeting method we also detected the *BCR::ABL1* fusion transcript in monocytes (Supplementary Fig. 21), an observation that was previously missed through FACS and FISH analysis.

In summary, these experiments showed that RoCK and ROI can be used both in cell lines and patient samples for the detection of both major and minor *BCR::ABL1* fusion transcripts in the context of cancer. Furthermore, thanks to the combination of WTA transcriptome information and target enrichment provided by RoCK and ROI, the *BCR::ABL1* fusion was detected in both the lymphoid precursor compartment as well as the myeloid lineage at the transcript level.

## Discussion

We present RoCK and ROI, a simple and highly versatile scRNA-seq-based method designed to capture specific transcripts (RoCKseq) and to selectively sequence regions of interest (ROIseq). RoCKseq works through the modification of standard barcoded beads, while ROIseq is mediated by addition of primers during library generation. We also provide a tailored data analysis workflow to systematically assimilate the targeted (TSO) and untargeted (WTA) reads. The full RoCK and ROI workflow, including protocols and data analysis, is freely available and can be performed in any laboratory with basic equipment, allowing researchers to target their transcripts of choice at will.

RoCK and ROI offers a rapid and reliable bead modification protocol (about 2 h, Supplementary Table 2) that is titratable and we show that it can be multiplexed using several RoCK captures and ROI primers. Furthermore, the bead modification is stable over months, allowing users working with time-sensitive material (for example clinical samples) to perform and validate RoCKseq bead modification prior to knowing the date of the future experiments. Multiplexing using large panels (with dozens or hundreds) of captures and/or ROI primers has not been tested. This setup would require substantial methodological development and extensive validation and optimization, including: iterative determination of optimal capture positions, target specificity as well as avoiding cross-reactivity between capture and ROI-primer sequences.

Compared to other targeted scRNA-seq methods (Table 1 and Supplementary Tables 2–9), RoCK and ROI offers the unique capability of combining targeted capture with targeted priming on the first strand level, while also profiling the standard transcriptome. Targeted capture at the RNA level (i.e. first step of library generation) has several advantages. First, it targets the RNA pool of interest at its native level. Second, it allows more flexibility during reverse transcription, e.g. gaining independence from 3′ biases derived from polyA capturing. In addition, intra-molecular priming for reverse transcription also allows detection of non poly-adenylated RNAs[33]. On the other hand, with targeted priming on the first strand level the process of directing reads to ROIs does not require additional PCR amplification steps for enrichment, other than the ones used in the standard BD protocol. In particular, 14 out of 18 current targeted scRNA-seq methods need to run additional amplification steps to enrich transcript or regions of interest (Table 1), a step which is known to introduce biases[12]. Sequencing-cost-wise, RoCK and ROI is standardly run with short-read sequencing, which is more affordable and thus more accessible compared to methods using long-read sequencing[30,36,40] (Supplementary

Tables 3–6). Out of the methods targeting at initial steps of library generation[20,24,28,33,40,58–60] (Table 1), RoCK and ROI is the only one combining both targeted capture and transcript targeting at the first strand level, making it unique and hence difficult to benchmark.

As we show in this manuscript, RoCK and ROI leads to the sequencing of off-targets. We believe this is due to the standard BD Rhapsody capture conditions used during the experiment (i.e., 2 min at room temperature using ice-cold buffer). These conditions suit the large diversity of transcripts differing in length, GC-content and sequence complexity and are optimized to capture RNAs via polyA capture. However, as a consequence, these relaxed RNA capture conditions eventually lead to the detection of non-polyadenylated transcripts via dT-capture, which may sum up to 20% of the detected transcripts in a standard experiment[61,62]. One possible explanation is that this may occur through the binding to internal polyA sequences present in transcripts[61]. Given that the (modified or unmodified) TSO oligos have higher melting temperatures ($T_m$: 63.5 °C) than the dT sequence (37.5 °C for a stretch of 25 dTs as present on BD Rhapsody beads), it is no surprise that a variety of non-targeted transcripts is recovered also in the TSO library by internal priming. In line with this reasoning, compared to WTA information, the TSO alignments are distributed across the entire transcript length (Fig. 5a). The value of this additional information has not been explored so far. At the molecular level, we believe (for both dT and even more for TSO-based capture) that the partial binding of a subset of nucleotides at the 3′ end of the capture oligo is sufficient to trap transcripts other than the targeted ones and that a perfect match of a few bases at the 3′ end can initiate reverse transcription. This is in line with the standard use of random hexamers for reverse transcription[63,64]. An improvement of the on-target RoCKseq capture (as well as ROIseq-based second strand synthesis) would require a change in the conditions of the standard capture and library preparation to increase the binding specificity of RoCK and ROI oligos to the targets. As a direct negative consequence, such changes (e.g., elevated temperatures or adapted buffers) would likely impair the dT-based capture of RNAs that occur simultaneously on the cell lysate.

Hence, while only a small proportion of reads from the TSO library are on target (Supplementary Fig. 14a–l), they suffice to reach more than a 20-fold enrichment (rockroi positive cell rate compared to the unmodified sample) resulting in up to 99% cells with detectable targeted transcript expression.

We have shown that RoCK and ROI is widely applicable and we firmly believe that this method is an important addition to the wealth of existing single cell transcriptome analysis protocols. Any change on the DNA level that is transcribed into RNA, polyadenylated or not, can be investigated. The list of genetic features that in our view can be analyzed is diverse and ranges from genetically engineered genes, inducible ectopic gene activation, transgenes, Cre-based recombination, naturally occurring sequence variations, CRISPR screens, or gene families, hence helping to explore and better understand complex biological systems in health and disease.

## Methods

### Design of capture sequences and fluorescent oligos

Detailed information on the design of ROIseq primers is available on protocols.io[65].

A list of primers used for the scRNA-seq experiments can be found in Supplementary Table 11 and Supplementary Data 2.

The sequence of the splint for the modification of TSO oligos was as follows: 5′ −24 nt coding sequence followed by a constant sequence-3′: 5′-NNNNNNNNNNNNNNNNNNNNNNNNNCATACCTACTACGCATA-3′ where the CATACCTACTACGCATA is the reverse complement of the TSO sequence. The sequence of the splint acts as a template for capture synthesis. For the modification of dT oligos on beads, the reverse complement of the TSO sequences was substituted by a polyA stretch

**Table 2 | Recommended conditions for the fluorescent assay to test modification of RoCKseq beads**

| Condition | Beads | Fluorescent oligo |
|---|---|---|
| Positive control dT | Barcoded beads (unmod) | polyA fluo oligo |
| Positive control TSO | Barcoded beads (unmod) | TSO fluo oligo |
| Negative control | Barcoded beads (unmod) | Fluo oligo for modification |
| RoCKseq beads | Barcoded beads (mod) | Fluo oligo for modification |
| dT control RoCKseq beads | Barcoded beads (mod) | polyA fluo oligo |
| Unmodified beads | Barcoded beads | – |

of 18 nts: 5′-NNNNNNNNNNNNNNNNNNNNNNNNNNAAAAAAAAAAAAAAAA AAAA-3′.

The polyA protective oligo used on the barcoded beads was 18 nucleotides in length: 5′-AAAAAAAAAAAAAAAAAA-3′.

The oligos were ordered in 0.2 μmol scale, HPLC grade, with 5′ phosphorylation. Before use, the oligos were resuspended in ddH$_2$O to generate a 100 μM stock solution.

Fluorescent oligos were designed by taking the first 20 nucleotides from the 5′ end of the splint. The fluorescent oligos were ordered in HPLC grade and in 0.2 μmol scale with a 5′ Atto647N modification and diluted in ddH$_2$O to generate a 100 μM stock solution.

### Protocol for polymerase-based bead modification for BD Rhapsody beads

A step-by-step protocol is available on protocols.io[65]. A general workflow is described in this section.

For modification of a full vial of BD Rhapsody beads (Enhanced Cell Capture Beads V2, Part Number 700034960, BD Rhapsody Enhanced Cartridge Reagent Kit, BD 664887), beads were split into four vials (500 μl per vial), which were processed in parallel. Beads were first washed once with Water Buffer (0.02% Tween 20 in ddH$_2$O) and once with TE/TW buffer (10 mM Tris, 1 mM EDTA, 0.02% Tween 20 in ddH$_2$O) using a magnetic rack for buffer exchange. Beads were next incubated with a T4 polymerase + splint mix (260 μl 5x T4 polymerase buffer (Thermo scientific EP0062), 130 μl 10 mM dNTPs (final concentration 1 mM), 857 μl ddH$_2$O, 6 μl of 1:1 mix of splint(s) and protective polyA oligo (final concentration of mix of splints and protective polyA 50 μM each); amounts for each of the four vials) without T4 polymerase for 5 min at 37 °C with shaking at 300 rpm in a thermomixer. Six μl of T4 DNA polymerase (5 U/μl units) was added to each vial and incubated for 10 min at room temperature on a MACSmix tube rotator (Miltenyi), followed by inactivation of the T4 polymerase by incubation for 10 min at 75 °C. The four bead vials were then washed as above with Water buffer followed by TE/TW buffer and incubated for 30 min 37 °C in a lambda exonuclease mix (95 μl 1x lambda exonuclease buffer (NEB M0262L), 832 μl ddH$_2$O, 21 μl lambda exonuclease enzyme (5000 U/mL); amounts for each of the four vials), followed by inactivation of the enzyme for 10 min at 75 °C. Beads were then washed again as above and stored at 4 °C in TE/TW buffer.

An aliquot (20 μl) was used for bead modification quality control. The T4 and lambda exonuclease reactions are also scaled down accordingly (T4 reaction: 40 μl T4 polymerase buffer, 20 μl 10 mM dNTPs (final concentration 1 mM dNTPs), 136 μl ddH$_2$O, 1 μl of polyA:splint(s) mix, 1 μl of T4 polymerase mix; lambda exonuclease reaction: 15 μl of lambda exonuclease buffer, 132 μl ddH$_2$O, 3 μl of lambda exonuclease enzyme).

### Protocol for fluorescent assay to quantify bead modification efficacy

A step-by-step protocol is available on protocols.io[65]. A general workflow is described in this section. To test RoCKseq bead modification, barcoded beads were incubated with multiple fluorescent oligos acting either as positive and negative controls or specific for the modification. RoCKseq modified beads and unmodified beads used for controls were incubated with fluorescent oligos for 30 min at 46 °C in BD Rhapsody lysis buffer (part number 650000064, BD Rhapsody Enhanced Cartridge Reagent Kit, BD 664887) with 1 M DTT (5.31 mM final concentration; part number 650000063,BD Rhapsody Enhanced Cartridge Reagent Kit, BD 664887).

Recommended conditions for the fluorescent assay are indicated in Table 2.

### Analysis of fluorescent signal from barcoded beads

The signal from barcoded beads after the fluorescent assay was measured at the Cytometry Facility at the University of Zürich using a FACS Canto II 2L with HTS (BD Biosciences, Switzerland). The signal from the Atto647N molecules was measured using the allophycocyanin (APC-A) channel. Gating for beads was performed on the forward scatter (FSC-A) vs. side scatter (SSC-A) scatterplot and 1000 beads per condition were measured. The .fcs files obtained from the analyser were imported into R (version 4.3.1) and plots were made primarily using the flowCore (version 2.14.0), flowViz (version 1.66.0), ggcyto (version 1.30.0) and ggplot2 (version 3.4.4) packages.

### ROIseq primer design

Detailed information on the design of ROIseq primers is available on protocols.io[65]. We recommend positioning the ROIseq primer between 50–400 bp upstream (e.g., 5′ and taking into account the transcript strand) of the RoCKseq capture. ROIseq primers were designed directly 5′ (max. 10 bp upstream) to the region of interest (ROI). The length of the primers we used is 12 nucleotides. Since 12 nucleotides will be included in the cDNA sequencing read (HTS), the ROIseq primer was designed in close proximity to the ROI. The ROIseq primer has the following structure: 5′-TCAGACGTGTGCTCTTCCGATCTNNNNNNNN NNNNN-3′; the N12 sequence of the ROIseq primer is identical to the coding strand. The primers were ordered from Microsynth in HPLC grade and at 0.2 μmol scale and resuspended in DNA Suspension buffer from Teknova (T0221).

### Library generation for RoCK and ROI

A step-by-step protocol is available on protocols.io[65]. A general workflow is described in this section. RNA capture and cDNA synthesis were performed following the manufacturer's instructions (Doc ID: 210966) using the following kits: BD Rhapsody Enhanced Cartridge Reagent Kit: BD 664887; BD Rhapsody Cartridge Kit: BD 633733; BD Rhapsody cDNA Kit: BD 633773; BD Rhapsody WTA Amplification Kit: BD 633801. To account for the bead loss during modification, RoCKseq beads were resuspended in 680 μl Sample Buffer (Cat. No. 650000062, BD Rhapsody Enhanced Cartridge Reagent Kit, BD 664887) instead of 750 μl.

The RoCK and ROI libraries were generated following the manufacturer's recommendations (Doc ID: 23-21711-00) with the following changes:

(1) Random priming and extension: ROIseq primers were added after beads were resuspended in Random Priming Mix. If a single ROIseq primer was added, 1 μl of the 100 μM primer was diluted 1:10 in ddH$_2$O and 4 μl of the diluted mix was added. If multiple ROIseq primers were used, 1 μl of each ROIseq primer (100 μM) was mixed, ddH$_2$O was added up to 10 μl and 4 μl of the diluted mix was added.

(2) RPE PCR: during RPE PCR, 1 µl of 100 µM T primer was added to each sample of RPE PCR Mix combined to Purified RPE product.

(3) Indexing PCR: for indexing of RoCKseq libraries, a separate PCR was performed substituting 5 µl of the Library Forward Primer with 5 µl of 100 µM of a custom indexing primer. The same primary library and reverse primers were used as recommended by the manufacturer.

If no ROIseq was being performed, points 2–3 were followed. A list of primers used for the scRNA-seq experiments can be found in Supplementary Table 11 and Supplementary Data 2. The T primer and Indexing primer were resuspended in DNA Suspension buffer from Teknova (T0221).

## Sequencing

Libraries were indexed using the BD Rhapsody Library Reverse primers as described by the manufacturer combined either with the BD Rhapsody Library Forward primer for the WTA-based information or the RoCKseq Indexing primer (see section "Library Generation for RoCK and ROI"). RoCKseq and WTA libraries of a given sample were indexed with the same 8 bp index sequence and pooled in a 1:1 concentration. For sequencing of pooled libraries including at least one RoCKseq modified sample (with or without ROIseq primers), a custom R1 primer was spiked in for the sequencing. Sequencing was performed at the Functional Genomics Centre Zurich (FGCZ) using a Novaseq 6000 and a full SP 200 flow cell for each experiment, except for the HeLa cell experiment which was sequenced on a Illumina NovaSeq X Plus using a 1.5 billion read flow cell. The length of R1 was 60 bp and the length of R2 was 62 bp for the cell line mixing experiments and *Pdgfrα*, while an R2 of 180 bp was used for the experiment on the leukemia cell lines and patient sample and the HeLa cell experiment. A 3% PhiX spike-in was used.

## Generation of stable cell lines

The FUGW plasmid (Addgene #14883) was used for the generation of the L-cells expressing eGFP. For the generation of the HEK293-T cells expressing tdTomato, the *eGFP* ORF in the FUGW plasmid was excised using the EcoRI and BamHI sites and substituted with the *tdTomato* sequence from the pCSCMV:tdTomato vector which was excised with the same restriction enzymes. The fluorescent cells were generated by lentiviral transduction. Lentiviruses were generated following the cultured Lipofectamine 3000 protocol supplied by the manufacturer. HEK293-T cells were in a T75 flask using 16 mL of packaging medium which was generated by mixing 47.5 mL Optimem reduced serum, 2.5 mL FBS, 100 µl sodium pyruvate and 500 µl Glutamax. On day 1, tube A was prepared by mixing 2 mL of Optimem with 55 µl of lipofectamine 3000. Tube B was prepared by mixing 2 mL Optimem, 17.8 µg of lentiviral packaging plasmid mix (4.8 µg pVSV-G, 9.6 µg pMDL, 3.4 µg pRev), 6 µg of the diluted plasmid containing *eGFP* or *tdTomato* and 47 µl of the P3000 reagent. Tube A and tube B were mixed and incubated at room temperature for 20 min. Four mL of medium was removed from the flask and substituted with the 4 mL of mix A and B. The cells were incubated for 6 h, after which the medium was removed and substituted with 16 mL packaging medium. On day 2, 24 h post transfection the volume of supernatant was collected and stored at 4 °C. The next day, 52 h post transfection the medium was collected and stored in the same Falcon tube as the day before. The medium was spun down 2000 rpm ( -859 g; Sorvall 75996445 rotor) for 3 min, after which it was filtered through a 45 µm filter into a new tube. The volume was then transferred to an Amicon tube and centrifuged 3000 × g for 10 min until the volume reached 500 µL on the Amicon tube. The liquid was then stored at −80 °C.

For lentivirus transduction, 300,000 HEK or L-cells were seeded onto a 6 well plate 12 h prior to transduction with 100 µl of concentrated viral supernatant in standard cell culture medium supplemented with 20 µg/mL polybrene (Sigma). The cells were then passaged 3 times.

To generate clonal cell lines, single cells were sorted into single wells of a 96-well plate. For the sorting of the cells, the cells were first of all dissociated with Trypsin-EDTA as described above. The cells were washed once with PBS and spun down at 290 × g for 5 min. A Zombie Violet viability staining was performed by resuspending the cells in 1 mL PBS and adding 2 µl of Zombie Violet (1:1000 dilution, Biolegend). The cells were then kept for 10 min in the dark, after which 9 mL of medium was added to quench the reaction. The cells were then spun down at 290 × g for 5 min, resuspended in 500 mL of medium and filtered through a Falcon 5 mL Round Bottom Polystyrene Test Tube with Cell Strainer Snap Cap (352235, Corning). Single cells were sorted in single wells of a 96-well plate at the Cytometry Facility at the University of Zürich using a BD S6 5 L cell sorter (BD Biosciences, Switzerland). The cell lines were then expanded and cultured as described above.

These cell lines are available upon request.

## Cell culture

L-cells and HEK293-T cells were ordered from ATCC (catalogue numbers: CRL-2648 for L-cells, CRL-3216 for HEK293-T). HeLa cells were kindly provided from the lab of Prof. Dr. Lukas Pelkmans, University of Zurich. The cell lines utilized in this study were obtained from established laboratory stocks. No additional authentication procedures, such as short tandem repeat (STR) profiling, karyotyping, DNA barcoding, or species-specific PCR assays, were performed on these cell lines prior to or during the course of the experiments. Therefore, the identity and purity of these cell lines were not independently verified for this study.

L-cells cells were cultured in Dulbecco's Modified Eagle Medium 1X (Thermo Fisher 41966-029) with 10% FBS (GIBCO, 10270-106) and 1% PIS in a 10 cm dish and maintained in an incubator at 37 °C and 5% $CO_2$. Cells were split at 80% confluence. To dissociate the cells, the medium was removed and 2 mL of Trypsin-EDTA (0.5%, no phenol red) were added to the dish, followed by 5 min in the incubator. The trypsin was inactivated by adding 8 mL of medium. To remove trypsin, cells were centrifuged for 5 min at 290 × g, the supernatant was removed and the pellet was resuspended in 10 mL of medium and plated depending on the wanted confluency. The total volume of the dish was 10 mL.

HEK293-T were also cultured in Dulbecco's Modified Eagle Medium 1X (Thermo Fisher 41966-029) with 10% FBS (GIBCO, 10270-106) and 1% PIS in a 10 cm dish and maintained in an incubator at 37 °C and 5% $CO_2$. Cells were split at 80% confluence. To dissociate the cells, the medium was removed and 2 mL of Trypsin-EDTA were added to the dish. The Trypsin-EDTA was immediately removed and the dish was placed in the incubator at 37 °C and 5% $CO_2$ for 1 min. Ten mL of medium was then added to the dish and the cell mixture was plated depending on the wanted confluency.

HeLa cells (gift from the lab of Prof. Dr. Lukas Pelkmans, University of Zurich) were cultured in Dulbecco's Modified Eagle Medium 1X (41965-039) with 10% FBS (GIBCO, 10270-106; no PIS) in a 10 cm dish and maintained in an incubator at 37 °C and 5% $CO_2$. Cells were split at 80% confluence. To dissociate the cells, the medium was removed and 2 mL of Trypsin-EDTA were added to the dish. The Trypsin-EDTA was immediately removed and the dish was placed in the incubator at 37 °C and 5% $CO_2$ for 1 min. Ten mL of medium was then added to the dish and the cell mixture was plated depending on the wanted confluency.

For the culture of cells for the *BCR::ABL1* experiment, SUP-B15, K562 and Loucy suspension cells were cultured and maintained in RPMI with 10% FBS and 1% PIS in cell culture flasks in an incubator at 37 °C and 5% $CO_2$.

**Table 3 | List of antibodies used for enriching non-leukemic live cells by FACS sorting**

| Antibody | Fluorochrome | µL | Dilution | Sorting strategy | Cell marker | Catalogue number | Company | Clone |
|---|---|---|---|---|---|---|---|---|
| CD45 | V500 | 5 | 1:20 | positive | lymphoid | 655873 | BD Horizon | 2D1 |
| CD235a | BV421 | 5 | 1:20 | negative | mature erythrocytes | 562938 | BD Horizon | GA-R2(HIR2) |
| Viability dye | FITC | 1 | 1:100 | negative | live cells | 130-135-318 | Miltenyi Biotec | |
| CD19 | PE-Cy7 | 5 | 1:20 | both | leukemia | IM3628 | Beckman Coulter | J3.119 |
| CD10 | APC-Fire750 | 1 | 1:100 | both | leukemia | 312229 | BioLegend | HI10a |
| CD13 | PE | 3.5 | 1:29 | Not for sorting | myeloid | 347406 | BD | L138 |
| CD33 | APC | 5 | 1:20 | Not for sorting | myeloid | 345800 | BD | P67.6 |
| CD34 | BV605 | 2 | 1:50 | Not for sorting | stem cells and precursor cells | 745247 | BD OptiBuild Biosciences | 8G12 |
| CD3 | BV711 | 6 | 1:17 | Not for sorting | T-cells | 344838 | BioLegend | SK7 |
| BSB | none | 10 | | For antibodies | – | | | |

## Preparation of single cell suspension from cell lines

Single cell solutions were prepared following the manufacturers recommendations. After dissociation and spinning down, the medium was removed and the cells were resuspended in 1 mL of Sample Buffer. The cells were filtered through a Round Bottom Polystyrene Test Tube with Cell Strainer Snap Cap (352235, Corning) and counted with a Neubauer chamber. The volume of cell solution to use was calculated using the following formula per manufacturers recommendations: (#cells in experiment x #samples x 1.36) / counted # of cells per µl. The calculated volume was then diluted to 650 µl of sample buffer before loading on the BD Rhapsody Express machine. For the mixing experiments, the same procedure as above was used and the same number of cells were mixed in a 1:1 ratio for a final volume of 1.3 mL.

For the generation of single cell suspension for the *BCR::ABl1* experiment, suspended cells were washed once with PBS and filtered through a 70 µm cell strainer. Cells were resuspended in PBS and counted with a Neubauer chamber. One million cells were resuspended in 1 mL of sample buffer and counted again at the BD Rhapsody Scanner. A ratio of 1:1:1 SUP-B15, K562 and Loucy cells was calculated using the BD Rhapsody Scanner and prepared before loading the sample on the BD Rhapsody Express machine.

## Isolation of cells from human sample

Bone marrow aspirates were enriched for mononuclear cells by Ficoll density gradient centrifugation prior to cell preservation by freezing. Frozen bone marrow samples were defrosted, washed twice in PBS and cells were counted with Trypan blue to assess sample quality. If the viability was below 75%, the cells underwent dead cell removal using magnetic bead-based dead cell removal (Dead Cell Removal Kit, Miltenyi Biotec) which was performed following the manufacturer's recommendations. Part of the sample was stained with fluorescent antibodies according to Table 3 for enriching non-leukemic live cells by FACS sorting.

Bone marrow sample was stained firstly with the viability dye for 10 min on ice, washed once with PBS and stained with the antibody mix for 30 min at room temperature in darkness. After staining the cells were washed once in PBS and resuspended in PBS for FACS sorting using the BD FACS Aria III Fusion. The cell sorting strategy is depicted in Supplementary Fig. 19b. Non-leukemic FACS sorted cells were mixed with the original untouched sample 2 to 1.

## Mice used in this study

Mice from the *Pdgfra*[H2BeGFP] strain[51] were purchased from Jackson Laboratories, United States of America (strain number 007669).

Mice in the *Pdgfra* scRNA-seq experiment were three males aged 2 months and 12 days (for two mice) and 1 month and 22 days.

Mice were housed in IVC (individually ventilated cages -T2 type, with plastic shelters inside, plus enrichment). The housing was under standard conditions, with 12 h day/night cycle. The temperature in the mouse house is kept at 22 °C and humidity at 50%.

## Sequencing of transgenic *Pdgfra* locus

To gain the sequence information for mapping of reads in the *Pdgfra*[H2BeGFP] strain, DNA from tail biopsies was PCR amplified using primers outside of the region removed during generation of the mouse strain[63], corresponding to a 6.5 kb fragment between BamHI-SmaI sites; sequences of primers forward: ACAGAGGCTGCCTCAAAGCTAG, reverse: CCATTGCCCAGATGGGAAGC) and cloned into pGEM-T easy vector (Promega). The insert was Sanger sequenced using M13 forward and reverse primers. Sequencing was performed by Microsynth.

## Colonic single cell isolation and cell sorting

Colonic tissues were obtained from *Pdgfra*[H2BeGFP] reporter mice[63]. The tissues were flushed with PBS, longitudinally opened and finely minced into 2 mm pieces. Minced tissue fragments were washed with PBS three times. Following the methodology outlined by Brügger et al.[52], tissue pieces underwent rounds of digestion to separate epithelial and mesenchymal fractions.

For the detachment of the epithelial fraction, the tissue pieces were incubated in Gentle Cell Dissociation Reagent (STEMCELL Technologies, Germany) while gently rocking for 30 min at room temperature. The pieces were pipetted up and down for the epithelial fraction to be detached. The epithelial fraction was then filtered through a Falcon 70-µm cell strainer (Corning, Switzerland), washed with plain ADMEM/F12 and incubated for 5 min at 37 °C in prewarmed TrypLE express (Gibco, Thermo Fisher, Switzerland). The gentleMACS Octo Dissociator (Miltenyi Biotec, Switzerland) m_intestine program was employed for single-cell dissociation. The obtained epithelial single-cell suspension was then filtered through a Falcon 40-µm cell strainer (Corning) and kept on ice in ADMEM/F12 supplemented with 10% FBS.

For dissociation of the mesenchymal fraction, the remaining tissue pieces (following epithelium detachment) were digested for 1 h at 37 °C under 110 rpm shaking conditions (gentleMACS) in DMEM supplemented with 2 mg/mL collagenase D (Roche) and 0.4 mg/mL Dispase (Gibco). The mesenchymal fraction was then filtered through a Falcon 70-µm cell strainer (Corning), washed with plain ADMEM/F12, and subsequently filtered through a Falcon 40-µm cell strainer (Corning).

The epithelial and mesenchymal cells were mixed and stained for 30 min on ice with anti-CD326(EpCAM)-PE-Cy5 (1:500, eBioscience/ Thermo Fisher, Switzerland, LOT 23658855, clone G8.8) in PBS. Prior to

cell sorting, all cells were stained for 5 min on ice with DAPI in PBS (1:1000, Thermo Fisher, Switzerland). Epithelial and mesenchymal cells labeled with PE-Cy5 and eGFP were sorted separately and subsequently mixed in a 1:1 ratio. Cells were sorted at the Cytometry Facility at the University of Zürich using a FACSAria III cell sorter (gates visible in corresponding figures) (BD Biosciences, Switzerland).

## RNAscope experimental procedure
The localisation of *eGFP* mRNAs in cells was performed with RNAscope (Advanced Cell Diagnostics, Germany) in a 96 well plate following the manufacturers recommendations (RNAscope Fluorescent Multiplex Assay). The fluorescent Probe - EGFP-O4 - Mycobacterium tuberculosis H37Rv plasmid pTYGi9 complete sequence (Advanced Cell Diagnostics, 538851) was used for all experiments. After DAPI staining, a protein stain was performed using Alexa Fluor 488 NHS-Ester (Succinimidylester) (Thermo Scientific, A20000). The wells were first of all washed with PBS, after which the supernatant was aspirated to 30 μl and 160 μl of CASE buffer (609.4 μl freshly thawed NaHCO$_3$, 15.63 μl Na$_2$CO$_3$, 2.5 mL of water) was added to each well and subsequently aspirated to 30 μL 0.5 μL of Alexa Fluor 488 NHS-Ester (Succinimidylester) were then added to the remaining CASE buffer and 30 μl of CASE stain were added to each well. The plate was incubated for 5 min at room temperature in the dark, followed by 4 washes with PBS.

## RNAscope image acquisition and analysis
Images were acquired using an automated spinning disk microscope Yokogawa CellVoyager 7000 equipped with a 60x water-immersion objective (1.4 NA, pixel size of 0.108 mm), 405/488/647 nm lasers, the corresponding emission filters and sCMOS cameras. Forty-five *z*-slices with 0.5 mm spacing were acquired per site. Image analysis was conducted with MATLAB (R2021b) and its image processing toolbox. Raw images were corrected for non-homogeneous illumination for each channel by dividing each pixel intensity value by its normalized value obtained from images of the corresponding fluorophores in solution. Cell segmentation was performed using the maximum-projected Succinimidyl ester staining channel and cellpose[66] using cyto2 model and a cell diameter of 200 pixels. Segmented cells touching image borders, smaller than $10^3$ or larger than $10^5$ pixels were discarded for further analysis. FISH channel was smoothed using a 3D Gaussian filter (s = 1 pixel) and FISH spots with *x* and *y* coordinates overlapping with segmented cells were detected in 3D using intensity thresholding (100 grays level value) followed by watershed segmentation with a minimum size of 9 voxels. Images were processed with ImageJ (Fiji version 2.0.0-rc-69/1.52p). Maximum intensity projections of 45 stacks are shown.

## BD Rhapsody barcode structure
Our data analysis workflow relies on mining the dual oligos present on beads from BD Rhapsody. Namely, whole transcriptome analysis (WTA) oligos profiling the non-targeted transcriptome have a tripartite cell barcode and a 8-nt-long UMI structure as follows: prepend-N{9}-GTGA-N{9}-GACA-N{9}-UMI with a prepend to choose from none, T, GT or TCA (Supplementary Fig. 20a). Template-switching oligos (TSO), modified via RoCKseq, are shaped N{9}-AATG-N{9}-CCAC-N{9}-UMI, without a prepend. The fixed parts between cell barcode 9-mers allow targeted (TSO) from untargeted (WTA) data to be distinguished.

## Single-cell data analysis workflow
We have developed a method to automate data processing from raw reads to count tables (and R SingleCellExperiment objects) and descriptive reports listing both on-target TSO (the targeted data) and off-target WTA (whole transcriptome analysis, the untargeted dT-captured RNAs) readouts. The software stack needed to run the method is provided via system calls (compiling recipes are provided), conda (environment files provided) or via Docker containers. The workflow is written in Snakemake[45] and hence easily deployable across

systems, including cloud provides, high performance and commodity computers.

To analyze their data, users need to provide their sequencing files in compressed FASTQ format (one file for the cell barcode plus UMI; and another for the cDNA) and a configuration file specifying the experimental characteristics and extra information, including:

- A genome (FASTA) to align the genome to (i.e., hg38, mm10 etc.). The genome needs to contain all (on target) captured sequences, so if these do not belong to the standard genome (i.e., *eGFP, tdTomato*), the genome FASTA file needs to be updated to append the extra sequences.

- Gene annotation (Gene Transfer Format; GTF) whose features are quantified separately for WTA and TSO. It is expected to contain a whole transcriptome gene annotation (i.e., Gencode, RefSeq etc) as well as an explicit definition of the RoCK and/or ROI targets captured by the TSO. Instructions to build this GTF are included within the software's documentation.

- A set of cell barcode whitelists following BD Rhapsody's standards (standard BD Rhapsody cell barcodes are included within the software).

- Parameters to fine tune CPU and memory usage.

The workflow (depicted in Fig. 2a) follows these steps:

(1) Index the reference genome with STAR[46].

(2) Subset reads match the WTA cell barcodes and map those to the transcriptome (genome plus GTF) using STARsolo[47]. Detected cell barcodes (cells) are filtered in at two levels: first, by matching to the user-provided cell barcode whitelist; and second, by applying the EmptyDrops[67] algorithm to discard empty droplets. We report two outputs from this step: the filtered-in cells according to the aforementioned filters; and the unbiased, whole-transcriptome WTA count table.

(3) Subset reads matching both the TSO cell barcode (CB) structure and the filtered in cell barcodes and map those to the transcriptome. Our reasoning is that the expected TSO transcriptional complexity is undefined and not usable to tell apart cells from empty droplets, so we borrow the filtered-in cells from the EmptyDrops results from the WTA analysis.

(4) (optional) Count on-target features in a more lenient way, filtering in multioverlapping and multimapping reads. This run mode is recommended when the captured regions target non unique loci (i.e., repetitive sequences).

Hence, our workflow always reports a WTA count table with as many genes as on-target and off-target gene features in the GTF, and per filtered-in cell barcode. As for the TSO, we offer these run modes:

- *tso_off_and_ontarget unique*: generates a count table for TSO reads from filtered-in cells; this count table has the same dimensions as the WTA.

- *tso_ontarget_multi*: creates a count table for TSO reads from filtered-in cells for only on-target features while allowing for and counting multioverlapping and multimapping alignments (downweighted to 1/(x*y) being 'x' the number of loci and 'y' the number of features each read aligns to and overlaps with, respectively).

- *all*: produces both 'tso off- and ontarget unique' and 'tso ontarget multi' outputs.

Finally, we generate an R SingleCellExperiment object with the aforementioned count tables and the following structure:

- *wta* assay: raw counts from the WTA analysis.

- (optional) *tso_off_and_ontarget_unique* assay: raw counts from the 'tso off- and ontarget' or 'all' run modes.

- (optional) *tso_ontarget_multi* altExp alternative experiment: raw counts from the 'tso ontarget multi' run mode. A complementary altExp built on WTA data, named 'wta_ontarget_multi', quantifies multi-overlapping and multimapping reads to the on-target regions in WTA data.

We also provide a simulation runmode to showcase the method, where raw reads (FASTQs), genome and GTF files are generated for three on-target features and one off-target feature across hundreds of cells before running the method.

The low-stringency annealing conditions for the ROI primer during second strand synthesis give rise to abundant mispriming events resulting in chimeric (artifact) cDNA reads (referred to as artifactually primed products, APPs) for both TSO off-targets and WTA. These APP cDNA reads start with the respective ROI sequence. The STAR alignment step mitigates this issue by soft-clipping the 5′ non-internal ROI sequence from cDNA reads in off-targets.

Our method is available at https://doi.org/10.5281/zenodo.11070200 under the GPLv3 terms.

### Detection of *BCR::ABL1* gene fusions

For the detection of the *BCR::ABL1* major and minor fusion transcripts, we first of all ran the single-cell data analysis workflow described above to retrieve valid cell barcodes using the WTA modality. Next, we designed a quality control quantification of APPs (Supplementary Fig. 20a), which harbor a $ROI^{BCR}$ primer sequence in their 5′ but align to *loci* other than *BCR* or *ABL1* (Supplementary Fig. 20b–d). Hence, we refine the analysis to increase quantification robustness of on-target (i.e. true *BCR::ABL1* gene fusions) and to discard library-induced APP artifacts (Supplementary Fig. 20a–e). For that, we trimmed the cDNA reads to remove 5′ sequences to the ROI primer while keeping the ROI primer sequence itself, and re-aligned these to ChimerDB 4.0[56] (a curated gene fusion database) using STAR with disabled splicing and soft-clipping (i.e., running an end-to-end mapping that ensured alignment in both the *BCR* and *ABL1* sides of the read). We reported alignments to minor and major *BCR::ABL1* fusions (as annotated by ChimerDBv4) on cDNA reads paired to valid (e.g., as in the WTA) cell barcodes and overlapping the fusion breakpoint (Fig. 7d–f, i–k and Supplementary Fig. 20f).

### Reference genomes, annotations and quantification strategy

To process the mouse and human mixing experiments, we generated a combined genome by concatenating GRCm38.p6 [https://www.ncbi.nlm.nih.gov/datasets/genome/GCF_000001635.26/] (mouse), GRCh38.p13 [https://www.ncbi.nlm.nih.gov/datasets/genome/GCF_000001405.39/] (human), *eGFP* (sequence obtained from FUGW Addgene #14883) and *tdTomato* (sequence obtained from pCSCMV:tdTomato Addgene #50530). For gene annotation, we used GENCODE's M25 (mouse) and v38 basic (human) and custom GTFs for *eGFP* and *tdTomato*. The data from the *Pdgfrα* experiment were mapped using the mouse genome GRCm38.p6 and GENCODE's M25 annotation, as well as the sequence for the *H2B-eGFP* construct in the transgenic mouse strain that was determined by sequencing the locus as described above. The data from the HeLa cell experiment was aligned to GRCh38.p13 (human), with Gencode's release 38 basic as annotation.

For mixing experiments, two regions were distinguished: the coding sequence (CDS) and the full transcripts (tx), the latter of which contains the 5′ and 3′ UTR in addition to the CDS.

GTF annotations are available under the GEO accession GSE266161.

Counts and UMIs of reads aligning to the WTA (untargeted) data (and typically corresponding to mouse or human genes) report unique mappers only. Counts and UMIs of reads aligning to targeted data (e.g. *eGFP*, *tdTomato*, or *Pdgfrα*) incorporate unique and also multimapping and multioverlapping reads, downweigthted accordingly (as reported in the *tso_ontarget_multi* usemode, section "Single-cell data analysis workflow"). We applied this counting strategy to circumvent the challenge posed by the repetitive nature of *tdTomato*, which is an almost perfect dimer (excluding the most 3′ and 5′ regions). For the HeLa cell experiment, only unique mappers were counted (including for the targets).

### Analysis of high-throughput sequencing data

**Software versions.** Data analysis was performed using R (version 4.3.2). Data wrangling was mainly performed using dplyr v1.1.4 and reshape2 v1.4.4. Plots were generated with ggplot2 v3.4.4 and ggrastr v1.0.2. Omics downstream analysis were run mainly using the Bioconductor ecosystem[17]: scran v1.30.2, scuttle v1.12.0, scDblFinder v1.16.0, Gviz v1.46.1, GenomicRanges v1.54.1, GenomicAlignments v1.38.2, GenomicFeatures v1.54.3 and edgeR v4.0.16. Alignment statistics were retrieved with Qualimap2 v2.3[68].

**Downsampling of single-cell data at the level of count tables.** When applicable (Supplementary Figs. 5d–f, 6d–f, 17a, b), data were downsampled across samples to the lowest average cell-wise library size using the *downsampleMatrix()* function of the scuttle package. The downsampled data were only used to generate quality control (QC) plots as well as calculating metrics such as mean number of genes or mitochondrial percent per cell.

Downsampling of single-cell data at the level of FASTQ files: to account for differences in read depth, the unmodified sample of the HeLa cell experiment using seqtk[69] (v 1.0-r82). To determine the ratio for downsampling, the total number of reads with valid barcodes from the rock_roi condition was divided by the number of reads with valid barcodes for the unmodified condition.

**Single-cell quality control metrics and filtering.** Quality control metrics for dT and TSO data such as percent mitochondrial transcripts, total number of genes and total number of transcripts were calculated using *addPerCellQCMetric()* from the scuttle package. Library size factors were calculated using *librarySizeFactors()* from the scuttle package.

Datasets were filtered for total number of UMIs and percent mitochondrial transcripts detected in the WTA data (first mixing experiment: unmod counts >3000, unmod_T counts >3700, mitochondrial counts for both samples >2% and <28%; second mixing experiment: unmod counts >3500, unmod_roi counts >3500, rock counts >2750, rockroi counts >3700, mitochondrial counts for both samples >2% and <28%; *Pdgfrα* experiment: for both samples genes >800, mitochondrial counts >1% and <75%; *BCR::ABL1* cell line experiment: for all cell lines genes >1000, mitochondrial counts >1% and <30%; *BCR::ABL1* patient sample: genes >200, mitochondrial counts >1% and <50%; HeLa cell experiment: for the unmodified sample counts >10,000 and mitochondrial counts >3% and <25%, for the downsampled unmodified sample, counts >15,000 and mitochondrial counts >6% and <25%, for the modified sample counts >10,000 and mitochondrial counts >6% and < 25%). If two species were present in the experiment (such as for the first and second mixing), the filtering was performed based on the sum of percent mitochondrial transcripts for the two species. Additionally, genes having less than three counts detected over all cells were filtered out in dT data.

Doublet removal was performed using scDblFinder[70] stratified by sample (e.g., rock, rockroi etc). Doublets were filtered out.

**Species assignment (mouse vs. human).** To distinguish between mouse and human cells in the two mixing experiments, we aligned the raw reads against a combined genome including mouse, human and other sequences (see section "Reference genomes and annotations"). Mouse cells were defined as having more than 50% counts to mouse genes or *eGFP* and *tdTomato* sequences and vice versa for human cells. Cells were labeled as unknown when having less than 50% of either mouse and human genes and were removed from the dataset for downstream analysis.

**Generation of coverage plots.** Coverage plots for the *eGFP* and *tdTomato* transcripts were generated using UMI-deduplicated Binary Alignment Map (BAM) files containing both unique and multimapping

alignments as generated by the workflow described above. The BAM files were split into mouse vs. human cells based on the species assignment described above. Plots were generated using the Gviz package[71]. Ranges for the annotation track were specified using the GenomicRanges and GenomicAlignments.

Coverage plots for ROIseq peaks in other genes were generated based on UMI-deduplicated bigWig files outputted by the workflow described above. Plots were generated using Gviz, as described above. The annotation track was generated by transforming the GTF used for mapping into a TxDb object using GenomicFeatures.

Coverage plots across mitochondrial transcripts were generated based on deduplicated bigWig files outputted by the automated pipeline described above. Plots were generated using Gviz as described above.

Coverage plots for the HeLa cell experiment were generated using UMI-deduplicated BAM files filtered for unique alignments. Plots were generated using Gviz as described above.

**Detection of positive cells (mixing mouse and human experiments).** The percent of positive cells for *eGFP* and *tdTomato* was based on counting UMI-deduplicated, unique or multimapping reads. The number of cells with non-zero counts for the CDS in the appropriate cell type (mouse or human cell line) was divided by the total number of cells after deduplication and filtering.

**Pseudobulk analysis of WTA signal across beads modifications.** For the *Pdgfrα* experiment rockroi vs. unmod analysis, to compare the WTA data between conditions, counts deriving from the previously filtered, doublet removed object were first aggregated by calculating the average logcount for each gene over each cluster. Genes with mean logcount across all cells (independent of cluster/condition) higher than 0.1 and variance higher than 0.5 were kept for the analysis. Bead modifications were compared by correlating (Pearson) pseudobulk values pairwise using the built-in *cor()* function from R.

For the two mixing experiments, counts were aggregated using the *aggregateAcrossCells()* function of the scuttle package. Genes with 0 counts across all samples were then removed from the dataset. LogCPM counts were calculated using the *cpm()* function from edgeR (*prior.count = 1*). Similarly, conditions were pairwise compared using Pearson correlation with the built-in *cor()* function from R.

For the comparison between the *Pdgfrα* rockroi unimodal and multimodal samples, datasets were subsetted for the same barcodes detected in both samples. Highly variable genes were calculated using the *modelGeneVar()* function (1938 genes with *p* value < 0.05). Counts per million were calculated using the *cpm()* function, after which data were subsetted based on the top 100 most highly expressed of the top 500 variable genes. The Pearson correlation was calculated using the built-in cor() function from R.

**Calculation of average *eGFP* counts in scRNA-seq experiments (RNAscope experiment).** The average *eGFP* counts detected in scRNA-seq experiments was calculated based on counting UMI-deduplicated alignments including multimappers. That is, reads aligning to *n* loci were assigned 1/*n* counts per locus. These values for the unmod and rockroi conditions were then divided by the sum of the mean RNAscope spots detected per cell for the two eGFP replicates divided by two ((131 + 118)/2).

**Gene-body coverage profile plots.** Data on the coverage along gene bodies (e.g., from transcription start site, TSS, to transcription end site, TES) were generated using rnaqc from Qualimap2[68]. Coverage data were imported into R and plotted using ggplot2.

**Gene biotypes analysis for TSO data.** Gene types detected in TSO data were derived by importing the GTF file used during mapping containing Gencode's assigned biotypes. The GTF was filtered for genes detected in the WTA from the previously QC-ed, doublet removed object.

**Sankey diagrams and number of reads and alignments.** Data plotted in Sankey diagrams were derived from BAM files generated by the automated pipeline described above. Data on counts (including on-target values) were generated in R. Sankey diagrams were plotted using SankeyMATIC (https://sankeymatic.com/, commit 088a339). The number of reads with canonical WTA and TSO barcode structure were calculated by running a regular expression on FASTQ files and without taking into account the variable regions whitelists. Sankey nodes reporting alignments or counts report our workflow's outputs, hence taking into account cell barcode whitelists and UMI duplicates. The number of alignments was extracted using bamqc from Qualimap2[68].

**Single-cell RNA-seq dimensionality reduction, embedding, and clustering for the *Pdgfrα* experiment.** Dimensionality reduction was performed using WTA-based data after quality control (including doublet removal). First of all the per-gene variance within each condition was modeled using the scran package (*modelGeneVar()* with condition id as block) on log-normalized counts (generated with the *logNormCounts()* from the scuttle package). Non-mitochondrial genes with biological variance larger than 0.01, *p* value smaller than 0.01 and mean normalized log-expression per gene were used for dimensionality reduction using the scran package. PCA was calculated with 30 components and used to build Uniform Manifold Approximation and Projection (UMAP) cell embeddings. Cells were clustered using *clusterCells()* from the scran package.

**Single-cell RNA-seq dimensionality reduction, embedding, and clustering for the *BCR::ABL1* cell line sample.** Dimensionality reduction was performed using WTA-based data after quality control (including doublet removal). First of all the per-gene variance within each condition was modeled using the scran package (*modelGeneVar()* with condition id as block) on log-normalized counts (generated with the *logNormCounts()* from the scuttle package). Non-mitochondrial genes with *p* value smaller than 0.5 were used for dimensionality reduction using the scran package. PCA was calculated with 30 components and used to build UMAP cell embeddings. Cells were clustered using *clusterCells()* from the scran package.

**Single-cell RNA-seq dimensionality reduction, embedding, and clustering for the *BCR::ABL1* patient sample.** Dimensionality reduction was performed with the Seurat[72] (version 5.1.0) and SeuratObject (version 5.0.2) package using WTA-based data after quality control (including doublet removal). The Matrix package was updated to version 1.6.4 for this analysis. A total of 2000 genes were used to find variable features using the *FindVariableFeatures()* function with the vst method. The *FindNeighbors()* function was run using 15 dimensions and the *RunUMAP()* function with 20 dimensions.

**Cell annotation (*Pdgfrα* experiment).** Clusters were manually annotated based on known cell markers in Supplementary Table 10. Cells were first of all split broadly into mesenchymal and epithelial and then clustered independently for annotation. Epithelial and mesenchymal clusters were defined as having mean logcounts per cell higher than 0.35 over all defined epithelial or mesenchymal markers respectively. Logcounts were calculated using the *logNormCounts()* function of the scuttle package. As one epithelial cluster had markers for both enteroendocrine and Tuft cells, the clustering was rerun on the subset of cells to distinguish the two cell types. Cells that were not classified as epithelial or mesenchymal were removed from the dataset.

**Junction analysis for *Pdgfrα*.** To detect reads spanning splice junctions, BAM files for WTA and TSO datasets were split by cell barcode and cell type (crypt top, crypt bottom and epithelial) and counted with featureCounts[73] specifying the *-J* (junction) flag and using fraction counts for multimappers (*--fraction*). Only canonical (annotated) splice junctions were kept into consideration. Only QC-filtered and doublet removed cell barcodes were included into the analysis.

The coverage, sashimi and alignment tracks for the roi_16 region were generated using Gviz. Only splice junctions with at least one UMI were filtered in.

We refer to GENCODE M25 ENSMUST00000202681.3 and ENSMUST00000201711.3 as short *Pdgfrα* transcripts; and to ENSMUST00000000476.14 and ENSMUST00000168162.4 as long transcripts.

### Statistics and reproducibility

No statistical methods were used to predetermine sample size. Experiments were all performed across thousands of cells. Instead of measuring replicates across samples, the method itself was validated across distinct cell lines, murine tissue and a human sample. The RNAscope experiment was performed on two replicates of eGFP-expressing cells. The experiments were not randomized. The investigators were not blinded to allocation during experiments and outcome assessment. Sex was not considered a factor in study design, as RoCK and ROI has no gender-specific components. The RoCK and ROI method was reproducible across samples and independent experiments. Low quality cells (determined based on detected number of genes, transcripts and mitochondrial content) were excluded after sequencing. The thresholds for exclusion were specific to the analyzed conditions and are specified in the "Methods" section. Measurements were taken across a substantial number of individual beads (approximately 1000 per condition) to assess variability in fluorescent signal between beads. In scRNA-seq experiments, multiple independent experiments were conducted with cell numbers consistent with standard usage on the BD Rhapsody platform. These experimental designs ensured sufficient cell numbers to robustly assess methodological performance, including variability at the single-cell level, while maintaining adequate read depth per cell.

### Ethics statement

The patient sample was pseudonymized and the patient gave written informed consent to scientific use of diagnostic leftover material. Ethical approval was obtained by the Ethics committee of Kiel University, Kiel, Germany (AZ: D416/21).

Mice: we affirm to have complied with all relevant ethical regulations for animal testing and research as follows. All animal based experimental procedures at the University of Zurich were performed in accordance with Swiss Federal regulations and approved by the Cantonal Veterinary Office (license ZH045/2019).

### Reporting summary

Further information on research design is available in the Nature Portfolio Reporting Summary linked to this article.

### Data availability

The raw and processed data generated in this study have been deposited in the GEO database under accession code GSE266161. The raw RNA-sequencing data for the patient sample have been deposited at the European Genome-phenome Archive (EGA), which is hosted by the EBI and the CRG, under accession code EGAS50000001366 (dataset: EGAD50000001976). Further information about EGA can be found at https://ega-archive.org and The European Genome-phenome Archive of human data consented for biomedical research. The processed data are deposited in the GEO database under accession code GSE266161. The RNAscope data generated in this study are provided in the Source Data

file. Reference genomes used in this work are available via accession codes GRCm38.p6 [https://www.ncbi.nlm.nih.gov/datasets/genome/GCF_000001635.26/] and GRCh38.p13 [https://www.ncbi.nlm.nih.gov/datasets/genome/GCF_000001405.39/]. Source data are provided with this paper.

### Code availability

Source code to analyze data from our method is available at https://zenodo.org/records/11070200 under the GPLv3 terms, and the code used to generate the figures and tables in this manuscript is available at https://zenodo.org/records/17356998 with the MIT license.

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

## Acknowledgements

We thank Vadir López-Salmerón, Cynthia Sakofsky, Hye-Won Song, Jannes Ulbrich, and Margaret Nakamoto from Becton Dickinson (BD) for their advice and technical support. We thank Catharine Fournier Aquino, Hubert Rehrauer, Andreia Cabral de Guevea, Hai Bui, and Joel Wirz at the Functional Genomics Center Zurich (FGCZ), Mario Wickert and Tatiane Gorski at the Cytometry Facility UZH, as well as Costanza Borrelli, Nidhi Agrawal, Jamie Little, Barbara Hochstrasser, and Reto Gerber for their technical support. We also thank George Hausmann for manuscript reading. We thank George Hausmann, Achim Weber, and Pierre-Luc Germain as well as the rest of the Robinson lab as well as the Basler lab for scientific discussions. We thank Fabienne Brutscher and Jamie Little for their support. Additionally, we are grateful for the reagents received from BD and the Pelkmans lab. Q.S. was supported by an EMBO post-doctoral fellowship (ALTF number: 170-2021) and a SNSF Swiss Post-doctoral Fellowships (TMPFP3_210503). T.V. was partially supported by the project National Institute for Cancer Research (Programme EXCELES, ID Project No. LX22NPO5102) funded by the European Union (Next Generation EU). This work was supported by the Swiss National Science Foundation (SNF), grant numbers 192475 (K.B.) and 310030_204869 (M.D.R.), and a grant from the Julius Klaus-Stiftung to E.B.

## Author contributions

E.B., K.B., and R.Z. conceived the study; E.B., K.B., I.M., and M.D.R. supervised the study; K.B. and E.B. were responsible for funding acquisition; E.B., G.M., I.M., M.D.R., K.B., R.Z., and A.E.M. contributed to experimental design; G.M. developed the wet-lab protocol, performed wet-lab experiments and downstream data analysis; G.M., E.B., M.J.B., L.B., and C.D.B. designed, performed and analyzed the *BCR::ABL1* experiments. I.M. developed the data analysis pipeline and performed downstream data analysis; M.D.R. performed downstream data analysis; M.D.B. performed the RNAscope experiment; H.F. performed the isolation of murine colonic cells for the scRNA-seq experiment; J.M. contributed to protocol development and validation; Q.S. performed the imaging and computational analysis of the RNAscope experiment; F.K. contributed to protocol validation; K.H. contributed to initial wet-lab protocol validation experiments; T.V. was responsible for mouse crosses, genotyping and licenses; G.M., E.B., I.M., and M.D.R. wrote the manuscript and all co-authors commented and edited it.

## Competing interests

K.B., E.B., G.M. as well as R.Z. and F.K. declare having received free-of-charge supplies from BD (Becton, Dickinson and Company). Other authors declare no competing interests. There is no IP filed on the work presented here.
