## [Transparent Peer Review file · Nature Communications]

RoCK and ROI: Single-cell transcriptomics with multiplexed enrichment of selected transcripts and region-specific sequencing

Corresponding Author: Dr Erich Brunner

A version of this paper was originally rejected for publication by Nature Communications, however that decision was reconsidered after appeal by the authors.

Version 2:

Reviewer comments:

Reviewer #1

(Remarks to the Author)

In this manuscript, Moro et describe the set up and application of “RoCK and ROI (Robust Capture of Key transcripts and Regions Of Interest)”, a scRNA-seq workflow that uses targeted capture to enrich for key transcripts, for example oncogenic fusion transcripts, thereby supporting the detection and identification of cell types and complex phenotypes in scRNAseq experiments. ROIseq directs a subset of reads to a specific region of interest via selective priming to ensure detection. The workflow was challenged to identify the most common BCR::ABL1 fusion transcripts and succeeded both in cell line mixtures and in a patient sample. Such an approach will be of great utility in scRNA-seq studies of chronic myeloid leukemia and Philadelphia chromosome-positive acute lymphoblastic leukemia.

I have the following remarks:

- Lines 440-441: “The position of the ABL1 fusion breakpoint is conserved across the major and the minor fusions and leads to the fusion of exon 2.” This is not completely correct. While the great majority of breakpoints involve fusion with exon 2 of ABL1, there are rare patients displaying atypical transcripts with fusions with exon 3 of ABL1. I would rephrase saying “The position of the ABL1 fusion breakpoint is relatively conserved across the major and the minor fusions and leads in nearly all cases to the fusion of exon 2.”

- Figure 7: a, Difference between major (e13a2 and e14a2) and minor fusions (e1a2) and RoCK and ROI targeting set up. This not exactly what panel a shows. Have legends for panels a and b been inadvertently swapped?

(Remarks on code availability)

Reviewer #2

(Remarks to the Author)

The manuscript “RoCK and ROI: Single-cell transcriptomics with multiplexed enrichment of selected transcripts and region-specific sequencing” by Moro, Mallona et al. reports on the development of a targeted single-cell RNA-seq method using modified BD Rhapsody beads. The protocol, coined ‘RoCK and ROI,’ targets both RNA and first-strand cDNA, achieving relevant enrichment rates in various cell types. The manuscript is well-written, and the figures nicely illustrate the results.

major comments:

1. It is unfortunate that the authors did not demonstrate the performance of their method by sequencing only the TSO libraries (WTA libraries were always included) and by using many more RoCK and matching ROI primers. As such, enrichment is only shown for one or a few targets per cell, while the method is in principle capable of targeted sequencing of dozens or hundreds of genes. It would be very informative if the authors could include such an experiment. I realize that the authors claim that it is a feature of the method that the classic transcriptome (WTA) is included, but for some applications, it is useful to only look at the genes of interest.

2. While the authors claim that low abundant genes can be targeted, I'm not convinced about this statement. First, no definition of 'low' is provided, it rather seems 'lower' than other/most genes detected in the regular workflow, but we know that this is only the tip of the iceberg. Not detected or low abundant in classic single cell RNA-seq does not automatically mean low. I propose that targeting a set of really low abundant genes (lower than 1 transcript per cell on average; e.g., 1 TPM down to, e.g., 0.01 TPM) and sequencing, only TSO libraries (see higher) can reveal the real low fraction of single-cell transcriptomes. The authors failed to demonstrate this power.
3. The detection of the BCR::ABL1 fusion in monocytes needs some further validation. First, can the authors demonstrate that the signal is higher than the background signal (e.g. 0.6% in Loucy for a given assay). Of note, also other cell clusters seem fusion positive (e.g. 2 top small clusters). Secondly, sorting monocytes and then doing wet lab validation could be an orthogonal validation.
4. The authors claim that an additional PCR step results in a bias (line 505). However, a) their protocol also includes PCR; b) UMLs largely correct for this bias, no?

minor comments:

1. The use of RoCK and ROI on the one hand and WTA/TSO or dT/TSO on the other hands does not help the reader. Could the nomenclature be standardized?
2. Is IP filed on the method? If yes, this should be mentioned in the conflict of interest section.
3. Why did the authors specifically use a T4 DNA polymerase for oligo extension instead of another DNA polymerase?
4. Did the authors explore thermal denaturation (and optional cleanup) to remove the splint oligo, instead of using an exonuclease treatment?
5. A Pearson correlation coefficient is not well suited to demonstrate concordance. The large dynamic range may give a false impression of good concordance. A scatterplot of fold changes vs fold changes (e.g. sample A vs B (or replicates A1 and A2) using 2 different methods in the x and y axis) or a Bland-Altman plot of log-transformed (normalized) counts is more sensitive to reveal differences.
6. What is the recommended (range of) distance(s) between the RoCK and ROI oligo(s)?
7. Lines 293-295: These detection rates depend on sequencing depth, right? How is it then possible to compare with previous reports?
8. Line 334: 80- and 94-fold increase, relative to what condition?
9. Line 404: The absence of Pdgfra reads in a cell that does not express Pdgfra is not evidence for the specificity. Rather, absence reads that have the Pdgfra targeting oligo but do not map to Pdgfra transcripts is supporting the selectivity of the method.
10. Line 532: 99% cell detection rate is not a very relevant metric, as it depends on sequencing depth and target abundance. Instead, enrichment is a better metric.
11. Lines 537-430: Could the method detect microRNAs and circular RNAs?
12. Non-affiliated persons should not use (registered) trademarks.
13. A space is needed between the value and the units (e.g. 75 °C, 100 µM, ...).
14. While there is a detailed protocol on protocols.io, some more details are needed in the methods section, such as end concentration of enzymes and oligonucleotides, custom sequencing primer sequence, ...
15. Line 592: Why is DTT added to the reaction?
16. "et al." does not contain the dot (.) everywhere
17. rpm values should be replaced by rcf or g units
18. Figure 2e (and others): are the units log2 scale?
19. Figure 4b/c: bars in bottom part of the plots are not clear. What do these mean?
20. Figure 4j: Why is the median value of the rockroi tx condition higher than rockroi CDS? RockROI is also only marginally better than RoCK. Could the authors comment?

(Remarks on code availability)

Reviewer #3

(Remarks to the Author)

The authors have developed a method for transcript enrichment and region-specific single cell sequencing by modifying the existing BD rhapsody single cell sequencing platform. They've tested the different aspects of their method on a technical level in addition to applying it in different biological settings.

In my opinion, the authors have tried to solve an underappreciated gap in current single cell RNA sequencing workflows, which are limited in their detection of both low abundant transcripts and good coverage of regions outside of the 3' and 5' transcript ends.

Additionally, I also commend the authors on their thorough testing and documentation of the protocols.

Although I am positive about the quality of the manuscript, I have a couple of points I would like to address.

Considering that this manuscript is focused on method development, the importance of this manuscript stands and falls with the ease of implementation of said method.

There are a number of improvements that should be made to ensure easier implementation and a higher adoption rate.

1. The protocol on protocols.io lacks molarities for common reagents such as Tris and EDTA, which makes reproducibility

challenging if not impossible.

2. Custom sequencing primer requirement. In the method section a custom R1 sequencing primer was mentioned. Custom sequencing primers will generally reduce the usability of the method as many core facilities and CROs do not have SOPs for such deviations. There are many sequencing adapters available in the standard illumina sequencing primer mixes, can the authors describe their choice for choosing to go with a custom primer option?

Additionally in Line 392 the following sentence is a bit confusing "This also removed the effect of the custom sequencing primer, which was only added in the cartridge with the multimodal condition" Can the authors rewrite this sentence to prevent confusion for future users of this method? I would suggest saying that the sequencing primer is only required for the multimodal libraries rather than talking about 'removing it's effect'.

3. I commend the authors for giving guidelines on how to design more successful oligos regarding the bead modifications. It is however still complex and laborious to perform both the modification and the testing. Can the authors provide success rates for their bead modifications when the suggested oligo design guidelines are followed? This will allow scientist without FACS access to still attempt the protocol, at an informed risk.

4. Have the authors tested if their data pipeline is deployable elsewhere? From the outside it looks like the authors have done their homework but I would still highly recommend them to ask scientists at other institutes on a different HPC to test the pipeline.

I have one request regarding the performed experiments

In supplemental figure 7 c-d the false positive rates of tdTomato and GFP are shown and are surprisingly low. This could be the result of having low expression levels of the fluorescent protein genes. Most of the use cases for a RoCK and ROI workflow would actually be on lowly expressed genes, however targeting mutations and/or splicing events on highly expressed genes would not be unrealistic.

If available, I would suggest using a highly expressed transcript to perform the false positive analysis on.

Overall I think this is a manuscript of value to scientists in both academia and industry and will open up scientific avenues in single cell sequencing that are currently inaccessible.

Dylan Mooijman

(Remarks on code availability)

Reviewer #4

(Remarks to the Author)

In the present manuscript, the authors describe a new method for single-cell RNA sequencing, called Robust Capture of Key transcripts and Regions Of Interest (RoCK and ROI). The method builds upon the addition of specific oligonucleotides to barcoded beads (BD Rhapsody barcoded beads from BD Biosciences), enabling the capture of predefined transcripts (ROI) in addition to the standard poly-T capture of transcripts containing poly-A tails. By this design, two separately indexed libraries are generated, subsequently pooled, and sequenced. The authors demonstrate that their method enables whole-transcriptome analysis (WTA) in combination with the targeted detection of specific genes of interest, such as reporter gene constructs, specific exon junctions, and gene fusions in leukemia samples. Importantly, the addition of specific oligos to the beads for targeted transcript capture apparently does not compromise WTA, which remains primed by the standard poly-T oligo attached to the beads. Furthermore, the authors describe a bioinformatic workflow developed to analyze the generated data.

This reviewer is primarily a user of various single-cell sequencing methods to address scientific questions rather than an expert in the technical details of method development. Therefore, my comments will focus on the method's applicability to users of single-cell sequencing techniques.

Undoubtedly, the described method has clear merits, as it enhances the ability to enrich for specific transcripts of interest and analyze their sequence information (e.g., presence of mutations and distinct mRNA isoforms) in combination with standard WTA. Thus, if this method is available to the broader scientific community, it would represent an important advancement. However, it is not entirely clear to this reviewer whether this is currently the case. Specifically, is it feasible for users to create custom-designed gel beads with specific oligos for capturing genes of interest, or is this a proprietary product intended for future commercialization by BD Biosciences? It would be helpful if the authors explicitly stated in the Discussion whether all reagents and protocols are available to the broader research community, as this would allow researchers to readily adopt the method, benchmark its performance against existing methodologies, and effectively apply the approach to answer relevant scientific questions. Additionally, it would be beneficial if the authors could comment on the potential to multiplex the method, thereby enabling simultaneous detection of multiple genes of interest.

(Remarks on code availability)

Version 3:

Reviewer comments:

Reviewer #2

(Remarks to the Author)

The authors have satisfactorily addressed all my comments and concerns.

(Remarks on code availability)

Reviewer #3

(Remarks to the Author)

My concerns have been addressed and I'm looking forward to seeing the manuscript published.

Dylan Mooijman

(Remarks on code availability)

Reviewer #4

(Remarks to the Author)

The authors have addressed all concerns raised in my previous review and I have no additional comments.

(Remarks on code availability)

Reviewer #1 (Remarks to the Author):

In this manuscript, Moro et describe the [setup] and application of “RoCK and ROI (Robust Capture of Key transcripts and Regions Of Interest)”, a scRNA-seq workflow that uses targeted capture to enrich for key transcripts, for example oncogenic fusion transcripts, thereby supporting the detection and identification of cell types and complex phenotypes in scRNAseq experiments. ROIseq directs a subset of reads to a specific region of interest via selective priming to ensure detection. The workflow was challenged to identify the most common BCR::ABL1 fusion transcripts and [succeeded] both in cell line mixtures and in a patient sample. Such an approach will be of great utility in scRNA-seq studies of chronic myeloid leukemia and Philadelphia chromosome-positive acute lymphoblastic leukemia.

I have the following remarks:

Lines 440-441: “The position of the ABL1 fusion breakpoint is conserved across the major and the minor fusions and leads to the fusion of exon 2.” This is not completely correct. While the great majority of breakpoints involve fusion with exon 2 of ABL1, there are rare patients displaying atypical transcripts with fusions with exon 3 of ABL1. I would rephrase saying “The position of the ABL1 fusion breakpoint is relatively conserved across the major and the minor fusions and leads in nearly all cases to the fusion of exon 2.”

We thank the review for the comment. We have changed the sentence accordingly and have additionally cited Burmeister et al., 2007¹.

Figure 7: a, Difference between major (e13a2 and e14a2) and minor fusions (e1a2) and RoCK and ROI targeting set up. This not exactly what panel a shows. Have legends for panels a and b been inadvertently swapped?

We thank the review for spotting the swapped panels. We have swapped panels a) and b) in Figure 7 (in accordance with the order of their appearance in the text).

Reviewer #2 (Remarks to the Author):

The manuscript “RoCK and ROI: Single-cell transcriptomics with multiplexed enrichment of selected transcripts and region-specific sequencing” by Moro, Mallona et al. reports on the development of a targeted single-cell RNA-seq method using modified BD Rhapsody beads. The protocol, coined ‘RoCK and ROI,’ targets both RNA and first-strand cDNA, achieving relevant enrichment rates in various cell types. The manuscript is well-written, and the figures nicely illustrate the results.

major comments:

It is unfortunate that the authors did not demonstrate the performance of their method by sequencing only the TSO libraries (WTA libraries were always included) and by using many more RoCK and matching ROI primers. As such, enrichment is only shown for one or a few targets per cell, while the method is in principle capable of targeted sequencing of dozens or hundreds of genes. It would be very informative if the authors could include such an experiment. I realize that the authors claim that it is a feature of the method that the classic transcriptome (WTA) is included, but for some applications, it is useful to only look at the genes of interest.

We thank the reviewer for the insightful comment and clarify that RoCK and ROI was specifically designed to enhance classical (untargeted) scRNA-seq techniques by preserving the whole transcriptome (WTA) information, ensuring compatibility with existing single-cell transcriptomic profiling techniques and data analysis workflows. We acknowledge the potential interest of TSO-only approaches with large panels of capture sequences and ROI primers. Nonetheless, these experiments require extensive validation and optimization. The key challenges for TSO-only experiments include:

1. Automated design of capture sequences and ROI primers is needed to ensure specificity and control cross-reactivity.
2. Iterative validation is essential to identify optimal capture positions. That is, capture sequences unable to trap their targets would be redesigned in subsequent iterations.
3. Library complexity limitations. TSO libraries lack barcode diversity insets (as shown in Supp Figure 1b), requiring high PhiX spike-ins and reducing efficiency. This caveat is circumvented by mixing TSO with WTA libraries, the latter having higher complexity and hence reducing PhiX requirements.

In summary, while RoCK and ROI could potentially support high-throughput targeted sequencing, establishing such a workflow would require substantial methodological development and lies beyond the scope of this manuscript.

To accommodate this point, we have added the following statement to the Discussion:

“Multiplexing using large panels (with dozens or hundreds) of captures and/or ROI primers has not been tested. This setup would require substantial methodological development and extensive validation and optimization, including: iterative determination of optimal capture positions, target specificity as well as avoiding cross-reactivity between capture and ROI-primer sequences.”

2. While the authors claim that low abundant genes can be targeted, I'm not convinced about this statement. First, no definition of 'low' is provided, it rather seems 'lower' than other/most genes detected in the regular workflow, but we know that this is only the tip of the iceberg. Not detected or low abundant in classic single cell RNA-seq does not automatically mean low. I propose that targeting a set of really low abundant genes (lower than 1 transcript per cell on average; e.g., 1 TPM down to, e.g., 0.01 TPM) and sequencing, only TSO libraries (see higher) can reveal the real low fraction of single-cell transcriptomes. The authors failed to demonstrate this power.

We thank the reviewer for this relevant comment. We agree that missing an expressed target in an scRNA-seq experiment may have various reasons, low expression levels just being one (and likely the most cited one in the field). Other parameters inherent to scRNA-seq steps, including PCR bias and random sampling, also influence the detection of expressed transcripts.

To look into this point, we first of all revisited our previous scRNA-seq experiments to determine the expression level of our targets compared to the rest of the genes in the dataset. As shown in figures provided below, our targets were between medium and highly expressed.

Mean CPMs across cells for unmodified versus modified sample for the Pdgfra experiment. Red dot indicates the targeted transcript. For the targeted transcripts, fractional multialigning counts were considered, while for all other genes (grey dots) only unique alignments were considered.

Mean CPMs across cells for unmodified versus modified sample for the second experiment performed on the mix of human and mouse cells. Red dot indicates the targeted transcripts. For the targeted transcripts, fractional multialigning counts were considered, while for all other genes (grey dots) only unique alignments were considered.

To address the very difficult task of: "[...] targeting a set of really low abundant genes (lower than 1 transcript per cell on average; e.g., 1 TPM down to, e.g., 0.01 TPM) and sequencing, only TSO libraries [...]", we set out to target a set of transcripts that are expressed in the range of a single transcript per cell on average, as suggested by the reviewer. Of note: the level of expression which we aimed to target is close to the detection limit (also considering signal to noise) of more sensitive methods, so expectations are very low. To explore the detection limit of RoCK and ROI, we performed an unmodified as well as a modified scRNA-seq experiment on HeLa cells, for which the expression levels of more than 900 transcripts had been determined by bDNA sm-FISH².

With this experiment we explored the lower limits of RoCK and ROI detection, finding that RoCK and ROI can - in some cases - slightly improve the detection also of very lowly expressed transcripts. To potentially improve successful targeting and to further enhance detection, multiple RoCK and ROI primer pairs would need to be tested to determine the optimal position for capturing and priming very rare targets. Importantly, we see that with the standard workflow (unmod) extremely lowly expressed genes can be detected (Supp Figure 11a-b, as in the case of HOXA1, detected as 0.0428 transcripts per cell on average between the two bDNA-smFISH replicates, expressed in less than 3% of cells according to bDNA-smFISH). This confirms that the expression level is not the only or main criteria for detectability as outlined above.

Mean CPMs across cells for unmodified versus modified sample for the HeLa cell experiment. Red dot indicates the targeted transcripts. Unique alignments were considered for all genes.

We have referenced this experiment in the main text and have additionally added an in-depth description as a Supplementary Note, as well as depicting the results in Supplementary Figures 9-12.

In addition, we have toned down the lowly expressed genes detection statements in the abstract taking the detection limits of RoCK and ROI into account.

3. The detection of the *BCR::ABL1* fusion in monocytes needs some further validation. First, can the authors demonstrate that the signal is higher than the background signal (e.g. 0.6% in Loucy for a given assay). Of note, also other cell clusters seem fusion positive (e.g. 2 top small clusters). Secondly, sorting monocytes and then doing wet lab validation could be an orthogonal validation.

Thank you very much for this valuable comment. Previous work from our department has indeed demonstrated the presence of *BCR::ABL1* in monocytes of patients with *BCR::ABL1*-positive acute lymphoblastic leukemia³. In that study, *BCR::ABL1* FISH performed on FACS-sorted diagnostic bone marrow samples confirmed the presence of the fusion in monocytes in 4 out of 16 analyzed cases. Of these, 5 cases were classified as exhibiting *BCR::ABL1* multilineage involvement, and 3 of these showed *BCR::ABL1* positivity in the monocyte compartment. Thus, it is not unexpected that our RoCK and ROI experiments also identified *BCR::ABL1*-positive monocytes. In the current analysis, *BCR::ABL1* was detected in 3.1% of monocytes. Given this frequency distribution close to the FISH sensitivity threshold and the expected loss of cells during FACS sorting, we believe that the suggested orthogonal validation would unfortunately not be feasible for this sample. However, a fusion positive rate of 3.1% still exceeds the false positive background (0.3%-0.6%) observed in our cell line experiments, supporting the conclusion that a small fraction of monocytes indeed harbors the *BCR::ABL1* fusion.

To further confirm that the *BCR::ABL1*-positive monocytes indeed belong to the assigned cell type, we compared the WTA from *BCR::ABL1*-positive monocytes (24 cells) to negative monocytes (756 cells). The WTA information was highly correlated, with a Pearson correlation of 0.828.

Pearson correlation on pseudobulks for BCR::ABL1-positive monocytes (x axis) and negative monocytes (y axis). Dots represent genes and pseudobulk indicate gene pseudobulks.

Additionally, both *BCR::ABL1*-positive and negative monocytes expressed monocyte markers but were negative for markers used for manual annotation of additional cell types (figures below, Supp Figure 21).

a) Heatmap of cell types and markers used for annotation, with monocytes split between BCR::ABL1-positive and negative; b) Heatmap of cell types and markers used for annotation including top 1'000 highly variable genes, with monocytes split between BCR::ABL1-positive and negative.

In terms of *BCR:ABL1* fusion detection at the alignment and counting level, we have updated the manuscript (main text and under Methods section “Detection of *BCR::ABL1* gene fusions”) and protocols.io (step 66) to clarify the data analysis procedure. We now explicitly discuss the impact of ROI primer design in downstream analysis, e.g., to detect true *BCR:ABL1* gene fusion transcripts while filtering out artificially primed products (APPs), which are cDNAs (cDNA reads) starting with the ROI^{BCR} primer (by-products of low ROI priming stringency).

4. The authors claim that an additional PCR step results in a bias (line 505). However, a) their protocol also includes PCR; b) UMIs largely correct for this bias, no?

We thank the reviewer for this comment, and we understand that the point needs clarification.

PCR-amplification is known to be biased, with short, GC-balanced cDNAs being preferentially amplified. These biases skew the representation of transcripts in the final dataset, with the loss of transcripts which are less favorable for amplification.

With our statement we wanted to emphasize that the targeting approach of RoCK and ROI does not involve additional PCR cycles other than the ones used in the standard BD Rhapsody protocol. In contrast, 14 out of 18 targeted scRNA-seq methods listed in Table 1 and the Supplementary Tables as well as the Discussion, require additional amplification steps for targeting increasing the mentioned biases.

To point b), UMIs can only correct for PCR bias if the targets are found in the final dataset, which is not always the case due to the biases mentioned above.

To increase clarity, we have changed the sentence at line 537 to the following:

“On the other hand, with targeted priming on the first strand level the process of directing reads to ROIs does not require additional PCR amplification steps for enrichment, other than the ones used in the standard BD protocol”.

minor comments:

1. The use of RoCK and ROI on the one hand and WTA/TSO or dT/TSO on the other hands does not help the reader. Could the nomenclature be standardized?

We agree that the nomenclature is demanding. We tried to streamline the nomenclature by exchanging the dT-based information by WTA. Currently, we use the following terms in the manuscript:

- WTA: Whole Transcriptome Analysis: data derived from libraries generated with dT oligonucleotides
- dT-based information: replaced by WTA information/libraries
- TSO information: data derived from libraries generated with TSO oligonucleotides
- dT oligonucleotides: refers to the actual dT oligonucleotides themselves (without generation of a WTA library)
- TSO oligonucleotides: refers to the TSO oligonucleotides themselves (without generation of a TSO library)

Finding an alternative solution to the RoCK and ROI terms is not trivial, as they represent two distinct aspects of our method.

2. Is IP filed on the method? If yes, this should be mentioned in the conflict of interest section.

There is no IP filed on the method, and we added this statement in the Competing Interests section for clarity. In line with the comment of Reviewer 4, we have additionally updated the Discussion section of the manuscript to clarify the method is free to use.

3. Why did the authors specifically use a T4 DNA polymerase for oligo extension instead of another DNA polymerase?

We thank the reviewer for this comment and acknowledge that this point was not specified in the manuscript. As the reaction needs to be inhibited after splint extension (T4 would start to attack single strands generated by the exonuclease), we discarded thermostable polymerases (such as Taq and Pfu polymerases). While we also considered Klenow fragment, we wanted the polymerase to possess 3' → 5' exonuclease activity for proofreading reasons. Additionally, T4 polymerases have a much lower error rate compared to Klenow fragments. Finally, the cost of T4 polymerases is also favorable compared to other polymerases.

We have added the following sentence to the manuscript:

“We chose this enzyme due to its low error rate and inactivation ability by heat, which is essential for subsequent steps in the bead modification protocol.”

4. Did the authors explore thermal denaturation (and optional cleanup) to remove the splint oligo, instead of using an exonuclease treatment?

We have tested both thermal denaturation and our lambda exonuclease enzymatic reaction for splint removal. As shown in Supp Figure 3a (and to the left), omission of the exonuclease treatment and of the exonuclease enzyme itself leads to a lower fluorescent signal for the modification compared to when the enzyme is present, indicating impaired binding of the fluorescent probe, corresponding to incomplete removal of the filled in strand. In both conditions, the T4 polymerase extension is followed by incubation at 75 °C for 10 minutes for enzyme inactivation, which also acts as a thermal denaturation step. In the condition with treatment but without exonuclease enzyme, a second 75 °C heating step (usually used for inactivation of the lambda exonuclease) was performed. This step was omitted in the condition without exonuclease treatment, where the beads were kept on ice instead of undergoing the bead modification. Of note, omission of the lambda exonuclease step and only including thermal denaturation also leads to a reduced binding of the fluorescent probe to the dT oligonucleotides in the fluorescent assay, indicating incomplete removal of the polyA protective oligonucleotide from the beads. In addition, the single nucleotides released by the lambda exonuclease treatment are easily removed in the

following wash step, while the complementary strand removed by heat treatment may be more difficult to remove completely (due to complementarity and re-annealing); residual oligonucleotides may thus impair subsequent reactions. For these reasons, we included the lambda exonuclease step for complementary strand removal.

5. A Pearson correlation coefficient is not well suited to demonstrate concordance. The large dynamic range may give a false impression of good concordance. A scatterplot of fold changes vs fold changes (e.g. sample A vs B (or replicates A1 and A2) using 2 different methods in the x and y axis) or a Bland-Altman plot of log-transformed (normalized) counts is more sensitive to reveal differences.

We agree and have added MA plots as Supplementary Figures (Supp Figure 5g-h, Supp Figure 7), a variant of Bland-Altman's particularly suited to RNA-seq data dynamic ranges, depicting the samples logratio vs average expression (log2 normalized counts).

6. What is the recommended (range of) distance(s) between the RoCK and ROI oligo(s)?

The recommended range of distances between the RoCKseq capture and ROIseq primers is between 50 and 400 bp distance from the RoCKseq capture. As an efficient reverse transcription reaction is needed for ROIseq binding, longer fragments may not be generated due to biases in this reaction. Various distances of ROIseq primers to RoCKseq captures were also validated in the *Pdgfra* experiment, in which 7 ROIseq primers differing in distance to the RoCKseq capture were used to direct information to exon-exon junctions. While recommendations for distances of ROIseq primers to RoCKseq captures were already present in the protocols.io and Material and Methods section of the paper, we have updated both to increase clarity in primer design.

We have updated the Material and Methods section of the paper with the following:

"We recommend positioning the ROIseq primer between 50-400 bp upstream (e.g., 5' and taking into account the transcript strand) of the RoCKseq capture"

We have also updated the protocols.io as follows:

"Vicinity to ROIseq primer: when performing RoCK and ROI, the RoCK capture should be chosen not more than 300 - 400 bp downstream or not less than 50 bp from the ROIseq primer. This accounts for the sequence on the bead (primer, barcode, UMI, TSO) added to the cDNA, as well as the optimal length for Illumina sequencing. Please note adaptors for sequencing add to the final product size as well."

7. Lines 293-295: These detection rates depend on sequencing depth, right? How is it then possible to compare with previous reports?

We fully agree with this statement. Additionally, the comparison of scRNA-seq data to RNAScope results also depends on the technology used. For example, HyPR-seq is based on probe-based capture at the RNA level, while RoCK and ROI uses targeted capture and priming on barcoded beads. While these types of experiments would be highly informative, to be properly controlled for they should be performed by

researchers that are experienced in both platforms, which is outside the scope of this paper. Additionally, targeting efficiency depends on the transcript itself and in particular on its sequence. While targeting efficiencies across platforms and experiments should thus not be compared, we believe that they offer a general indication of the range of improvement.

8. Line 334: 80- and 94-fold increase, relative to what condition?

We agree with the reviewer that this point was unclear.

We have amended the manuscript as following:

“This was especially true for the rock and rockroi conditions, where an 80-fold and 94-fold increase in on-target (unique or not) alignments in rock and rockroi was observed, respectively, in the TSO data compared to WTA (Supp Figure 14j and Supp Figure 14l).”

9. Line 404: The absence of Pdgfra reads in a cell that does not express Pdgfra is not evidence for the specificity. Rather, absence reads that have the Pdgfra targeting oligo but do not map to Pdgfra transcripts is supporting the selectivity of the method.

We thank the reviewer for this comment. For clarity, we have amended the statement to:

“As expected, these reads were detected exclusively in fibroblast clusters.”

without any mention of the RoCK and ROI specificity.

10. Line 532: 99% cell detection rate is not a very relevant metric, as it depends on sequencing depth and target abundance. Instead, enrichment is a better metric.

We have added the suggested metric to the text:

“Hence, while only a small proportion of reads from the TSO library are on target (Supp Figure 14a-l), they suffice to reach more than a 20-fold enrichment (rockroi positive cell rate compared to the unmodified sample) resulting in up to 99% cells with detectable targeted transcript expression.”

11. Lines 537-430: Could the method detect microRNAs and circular RNAs?

We have not tested circular RNA detection, but we believe that circular RNAs could be targeted with our method, as they are single-stranded and can thus be captured on the beads. Additionally, their median length is around 530 bp⁴, long enough to combine both RoCKseq capture with ROIseq priming. Circular RNAs are standardly detected in scRNA-seq experiments⁵ most likely due to internal dT-capture as they are not polyadenylated. RoCK and ROI could be used to enhance the detection of circular RNAs of interest. Our Snakemake workflow does not provide built-in circular RNA detection algorithms.

We have not attempted to detect microRNAs either. The length of mature microRNAs (miRNAs) is between 19-25 bp⁶, comparable to the length of the capture sequence on the beads. Shorter captures may be explored to capture mature microRNAs. However, the resulting cDNA including bead-based information and the targeted mature microRNA would likely be removed during the cleanup with size selection beads due to length restriction. For this reason, we believe that their detection would be problematic with RoCK and ROI. Our Snakemake workflow does not provide built-in microRNA detection tools.

Precursors of microRNAs such as pri-miRNAs and pre-miRNAs are up to a few kilobases⁷ in length and therefore could be detected using an appropriate capture and ROI primer pair that allows to sequence through the part encoding for the miRNA. As both pre-miRNAs and pri-mRNAs have very strong secondary structures, capture and ROI-primer binding may be impaired. Additionally, while pre-miRNAs are exported to the cytoplasm for processing, the presence of pri-miRNAs is restricted to the nucleus, further impacting their detection in standard scRNA-seq experiments. For this reason, we believe that studying miRNAs remains challenging with RoCK and ROI.

Given the absence of empirical data on microRNAs or circular RNAs, we have not updated the manuscript.

12. Non-affiliated persons should not use (registered) trademarks.

We thank the reviewer for noticing that the TM trademark was present in the Methods section. We have removed all occurrences in the text.

13. A space is needed between the value and the units (e.g. 75 °C, 100 µM, ...).

We have updated the manuscript to add spaces between values and units.

14. While there is a detailed protocol on protocols.io, some more details are needed in the methods section, such as end concentration of enzymes and oligonucleotides, custom sequencing primer sequence, ...

We thank the reviewer for this comment and have updated the Methods section of the manuscript with information on the molarities and concentrations of buffers and components of the reactions used for bead modification and the fluorescent assay (under “Protocol for polymerase-based bead modification for BD Rhapsody beads“ and “Protocol for fluorescent assay to quantify bead modification efficacy by FACS analysis”). We have also added this information to protocols.io. The custom sequencing primer is included in Supplementary Table 11.

15. Line 592: Why is DTT added to the reaction?

This is a recommendation made by the company (Becton Dickinson) and the reaction has not been tested without. We use the same lysis buffer for downstream reactions using modified beads, as suggested by the vendor. Since it does not change bead integrity, we decided to use it as suggested also for this assay.

16. “et al.” does not contain the dot (.) everywhere

We thank the reviewer for noticing that the dot was missing in some of the et al. We have updated the manuscript accordingly.

17. rpm values should be replaced by rcf or g units

We have added g units in the manuscript where appropriate, mentioning also the equipment used.

18. Figure 2e (and others): are the units log2 scale?

We thank the reviewer for this comment. We have added scaling of the axis to each axis label. We have updated the plot axes labels to clarify they are log2-transformed CPMs.

19. Figure 4b/c: bars in bottom part of the plots are not clear. What do these mean?

We have updated the figure by simplifying the annotations. Additionally, we have added the following indications in the legend for Figure 4:

“Bars at the bottom of panels (c) and (d) indicate regions in the eGFP and tdTomato transcript and CDS, also used for counting. The CDS is indicated in green or magenta, respectively, while the UTRs are indicated in white.”

We have also amended the legend for Supp Figure 8:

“a-b, Zoom in of 3' UTR from coverage plot in Figure 4b (a) and Figure 4c (b). The white bar indicates the zoomed in 3' UTR.”

20. Figure 4j: Why is the median value of the rockroi tx condition higher than rockroi CDS? RockROI is also only marginally better than RoCK. Could the authors comment?

We thank the reviewer for this comment. The signal from the transcript (tx) will always be higher compared to the one from the CDS, as we define the transcript as CDS plus UTRs. We agree that the rockroi condition is only marginally better than the rock only condition. While the aim of RoCKseq is an increase in transcript detection, the aim of ROIseq is to direct the information to regions of interest, after the transcript is captured. As noted in the manuscript, in most cases ROIseq can only be performed in combination with RoCKseq (while the opposite is not true). The purpose of ROI primers is to increase the information at a specific position of the transcript, and thus the overall increase in information will be lower compared to RoCKseq, in which the signal is dispersed across a wider transcript region due to the randomeres generating different 5' cDNA ends.

We have updated the glossary as follows:

“tx: transcript, including CDS and UTRs.”

In addition, a more detailed explanation is present in the Materials and Methods section, paragraph

“For mixing experiments, two regions were distinguished: the coding sequence (CDS) and the full transcripts (tx), the latter of which contains the 5’ and 3’ UTR in addition to the CDS. [...]”

Reviewer #3 (Remarks to the Author):

The authors have developed a method for transcript enrichment and region-specific single cell sequencing by modifying the existing BD rhapsody single cell sequencing platform. They’ve tested the different aspects of their method on a technical level in addition to applying it in different biological settings.

In my opinion, the authors have tried to solve an underappreciated gap in current single cell RNA sequencing workflows, which are limited in their detection of both low abundant transcripts and good coverage of regions outside of the 3’ and 5’ transcript ends.

Additionally, I also commend the authors on their thorough testing and documentation of the protocols. Although I am positive about the quality of the manuscript, I have a couple of points I would like to address. Considering that this manuscript is focused on method development, the importance of this manuscript stands and falls with the ease of implementation of said method.

There are a number of improvements that should be made to ensure easier implementation and a higher adoption rate.

1. The protocol on protocols.io lacks molarities for common reagents such as Tris and EDTA, which makes reproducibility challenging if not impossible.

Thank you for pointing this out. We have added the relevant information to protocols.io in the “Materials” section and directly in the “Steps” section.

2. Custom sequencing primer requirement. In the method section a custom R1 sequencing primer was mentioned. Custom sequencing primers will generally reduce the usability of the method as many core facilities and CROs do not have SOPs for such deviations. There are many sequencing adapters available in the standard illumina sequencing primer mixes, can the authors describe their choice for choosing to go with a custom primer option?

We thank the reviewer for pointing out this important addition to the RoCK and ROI protocol and are aware that the use of a custom sequencing primer is supported differently depending on the sequencing facility. This issue has also been brought up frequently by RoCK and ROI users. The custom primer is complementary to the primer on the TSO oligonucleotides (Supp Figure 1a-b, T primer, which differs from the one on the dT

oligonucleotides called U primer) and is added to obtain information from the oligonucleotides. None of the Illumina primers used for sequencing are complementary to the T primer and for this reason the custom sequencing primer needs to be supplied. During method development, we considered appending a Universal Primer adapter to the TSO oligonucleotides during indexing PCR (the primer would thus contain Illumina adapter – U sequence - T primer sequence). We chose against this set up as all R1 reads would start with the T primer sequence before entering the barcode region, increasing the read length to the detriment of the sequencing cost. In addition, appending a universal primer may require a higher PhiX spike-

in as the TSO oligos do not contain staggered ends. However, this could be a viable solution for users in which the sequencing facility does not support the use of custom sequencing primers.

We have updated the protocols.io as follows:

“As a replacement for the custom sequencing primer (which we have not tested), indexing can be performed with a forward primer composed of the 5'-Illumina sequencing adapter + U primer + T primer, thus appending a U primer sequence to the construct. In this case, all R1s would start with a T primer before entering the barcode region. We caution users against this set up, as a longer read length is necessary, which is accompanied by higher cost. Additionally, this may require a higher PhiX spike-in percentage. This may however be a viable solution for users who cannot use a custom primer for sequencing.”

Additionally in Line 392 the following sentence is a bit confusing “This also removed the effect of the custom sequencing primer, which was only added in the cartridge with the multimodal condition” Can the authors rewrite this sentence to prevent confusion for future users of this method? I would suggest saying that the sequencing primer is only required for the multimodal libraries rather than talking about ‘removing it’s effect’.

We agree that the formulation may be misleading and have thus changed the sentence to enhance understanding:

“As the custom sequencing primer is used to obtain information on the TSO libraries, it was not added to the cartridge in which exclusively WTA profiles were recorded.”

3. I commend the authors for giving guidelines on how to design more successful oligos regarding the bead modifications. It is however still complex and laborious to perform both the modification and the testing. Can the authors provide success rates for their bead modifications when the suggested oligo design guidelines are followed? This will allow scientist without FACS access to still attempt the protocol, at an informed risk.

In our hands, when following the specified guidelines outlined on protocols.io, the bead modification success rate is 100%. We do however still recommend using the FACS assay before performing the scRNA-seq experiment, as the reagents used for the bead modification itself may have gone bad, impacting the success of the modification itself.

We have updated the protocols.io with additional information for splint design, as well as a recommendation to perform the fluorescent assay before going into the scRNA-seq experiment (whenever possible).

4. Have the authors tested if their data pipeline is deployable elsewhere? From the outside it looks like the authors have done their homework but I would still highly recommend them to ask scientists at other institutes on a different HPC to test the pipeline.

Our workflow is written in Snakemake and also automates the software environments installation in a reproducible manner (using conda or aptainer). Hence, we benefit from Snakemake’s flexibility in execution, which, in principle, covers Slurm, Grid, htcondor etc, as well as several cloud providers (AWS, azure and so on), as listed in <https://snakemake.github.io/snakemake-plugin-catalog/>.

One of our users (unrelated to the method development team and from an external university, the Medical University of Innsbruck) has run our workflow in their local HPC premises using Slurm. We have not tested distributed nor cloud computing ourselves.

We have updated the manuscript `Single-cell data analysis workflow` methods section to describe deployment and execution.

I have one request regarding the performed experiments

In supplemental figure 7 c-d the false positive rates of tdTomato and GFP are shown and are surprisingly low. This could be the result of having low expression levels of the fluorescent protein genes. Most of the use cases for a RoCK and ROI workflow would actually be on lowly expressed genes, however targeting mutations and/or splicing events on highly expressed genes would not be unrealistic.

If available, I would suggest using a highly expressed transcript to perform the false positive analysis on.

We agree that this is an interesting point. We have noted that the false positive rate is in a similar range for all three of the performed experiments (targeting of *eGFP*, *tdTomato*, *Pdgfra* and *BCR::ABL1* fusions), which are expressed at different levels (plots depicting the baseline expression level as well as the increase using our method are part of the reply to reviewer 2) Although none of them are highly expressed, this indication speaks towards the fact that the false positive rate is independent of the targeted transcript itself, as well as of the level of expression of such transcript.

Overall I think this is a manuscript of value to scientists in both academia and industry and will open up scientific avenues in single cell sequencing that are currently inaccessible.

Thank you for the positive feedback.

Dylan Mooijman

Reviewer #4 (Remarks to the Author):

In the present manuscript, the authors describe a new method for single-cell RNA sequencing, called Robust Capture of Key transcripts and Regions Of Interest (RoCK and ROI). The method builds upon the addition of specific oligonucleotides to barcoded beads (BD Rhapsody barcoded beads from BD Biosciences), enabling the capture of predefined transcripts (ROI) in addition to the standard poly-T capture of transcripts containing poly-A tails. By this design, two separately indexed libraries are generated, subsequently pooled, and sequenced. The authors demonstrate that their method enables whole-transcriptome analysis (WTA) in combination with the targeted detection of specific genes of interest, such as reporter gene constructs, specific exon junctions, and gene fusions in leukemia samples. Importantly, the addition of specific oligos to the beads for targeted transcript capture apparently does not compromise WTA, which remains primed by the standard poly-T oligo attached to the beads. Furthermore, the authors describe a bioinformatic workflow developed to analyze the generated data.

This reviewer is primarily a user of various single-cell sequencing methods to address scientific questions rather than an expert in the technical details of method development. Therefore, my comments will focus on the method's applicability to users of single-cell sequencing techniques.

Undoubtedly, the described method has clear merits, as it enhances the ability to enrich for specific transcripts of interest and analyze their sequence information (e.g., presence of mutations and distinct mRNA isoforms) in combination with standard WTA. Thus, if this method is available to the broader scientific community, it would represent an important advancement. However, it is not entirely clear to this reviewer whether this is currently the case. Specifically, is it feasible for users to create custom-designed gel beads with specific oligos for capturing genes of interest, or is this a proprietary product intended for future commercialization by BD Biosciences? It would be helpful if the authors explicitly stated in the Discussion whether all reagents and protocols are available to the broader research community, as this would allow researchers to readily adopt the method, benchmark its performance against existing methodologies, and effectively apply the approach to answer relevant scientific questions.

We thank the reviewer for the comment and acknowledge that the open availability of the method to the scientific community was not emphasized strongly enough in the manuscript. We have added the following sentence to the Discussion:

“The full RoCK and ROI workflow, including protocols and data analysis, is freely available and can be performed in any laboratory with basic equipment, allowing researchers to target their transcripts of choice at will.”

Additionally, it would be beneficial if the authors could comment on the potential to multiplex the method, thereby enabling simultaneous detection of multiple genes of interest.

We thank the reviewer for this comment. In this manuscript, we present the multiplexing of three RoCK captures and eight ROI primers. Scaling up the number of capture sequences and ROI primers to large panels is of high interest but requires careful method development paired with extensive optimization and validation of the experimental steps.

An automated design of the oligonucleotide panels is a prerequisite, considering target specificity, and avoiding cross-reactivity between capture sequences on the beads. In addition, the optimal capture position for each transcript and ROI primers designed by the pipeline needs to be experimentally validated in an iterative approach. This iteration requires larger amounts of a well-defined test sample where expression of every member of the panel is guaranteed and stable. Capture sequences incapable of trapping the respective target would require a re-design in the next round. Taken together, scaling up the number of capture sequences and ROI primers to large panels is a long and complex process, with similar efforts to the actual design of the RoCK and ROI method itself.

We have added the following statement to the Discussion:

“Multiplexing using large panels (with dozens or hundreds) of captures and/or ROI primers has not been tested. This setup would require substantial methodological development and extensive validation and optimization, including: iterative determination of optimal capture positions, target specificity as well as avoiding cross-reactivity between capture and ROI-primer sequences.”

Of note: A similar point has been raised by reviewer 2 (remark 1).

References

1. Burmeister, T. et al. Atypical BCR-ABL mRNA transcripts in adult Acute lymphoblastic leukemia. *Haematologica* 92, 1699–1702 (2007).
2. Battich, N., Stoeger, T. & Pelkmans, L. Image-based transcriptomics in thousands of single human cells at single-molecule resolution. *Nat Methods* 10, 1127–1133 (2013).
3. Nagel, I. *et al.* Hematopoietic stem cell involvement in BCR-ABL1–positive ALL as a potential mechanism of resistance to blinatumomab therapy. *Blood* 130, 2027–2031 (2017).
4. Ding, X. et al. Profiling expression of coding genes, long noncoding RNA, and circular RNA in lung adenocarcinoma by ribosomal RNA -depleted RNA sequencing. *FEBS Open Bio* 8, 544–555 (2018).
5. Wu, W., Zhang, J., Cao, X., Cai, Z. & Zhao, F. Exploring the cellular landscape of circular RNAs using full-length single-cell RNA sequencing. *Nat Commun* 13, 3242 (2022).
6. Ranganathan, K. & Sivasankar, V. MicroRNAs - Biology and clinical applications. *J Oral Maxillofac Pathol* 18, 229 (2014).
7. Jin, W., Wang, J., Liu, C.-P., Wang, H.-W. & Xu, R.-M. Structural Basis for pri-miRNA Recognition by Drosha. *Molecular Cell* 78, 423-433.e5 (2020).